# Live Graph Lab: Towards Open, Dynamic and Real Transaction Graphs with NFT

**Zhen Zhang, Bingqiao Luo, Shengliang Lu, Bingsheng He**
National University of Singapore
zhen@nus.edu.sg,luo.bingqiao@u.nus.edu,lusl@nus.edu.sg,hebs@comp.nus.edu.sg

## Abstract

Numerous studies have been conducted to investigate the properties of large-scale temporal graphs. Despite the ubiquity of these graphs in real-world scenarios, it's usually impractical for us to obtain the whole real-time graphs due to privacy concerns and technical limitations. In this paper, we introduce the concept of *Live Graph Lab* for temporal graphs, which enables open, dynamic and real transaction graphs from blockchains. Among them, Non-fungible tokens (NFTs) have become one of the most prominent parts of blockchain over the past several years. With more than $40 billion market capitalization, this decentralized ecosystem produces massive, anonymous and real transaction activities, which naturally forms a complicated transaction network. However, there is limited understanding about the characteristics of this emerging NFT ecosystem from a temporal graph analysis perspective. To mitigate this gap, we instantiate a live graph with NFT transaction network and investigate its dynamics to provide new observations and insights. Specifically, through downloading and parsing the NFT transaction activities, we obtain a temporal graph with more than 4.5 million nodes and 124 million edges. Then, a series of measurements are presented to understand the properties of the NFT ecosystem. Through comparisons with social, citation, and web networks, our analyses give intriguing findings and point out potential directions for future exploration. Finally, we also study machine learning models in this live graph to enrich the current datasets and provide new opportunities for the graph community. The source codes and dataset are available at https://livegraphlab.github.io.

## 1 Introduction

Temporal graphs provide an accurate representation of real-world systems, including social networks, transaction networks and the Web [5, 38, 39, 41], etc. By investigating temporal graphs, we can gain insights into the temporal dynamics and understand how these systems evolve and function [86, 6]. Notably, a growing number of graph mining algorithms [36, 25, 85] and graph systems [69, 35, 13] have been developed. However, with this continuously growing trend, several severe issues emerge, which limit the further development of graph community. In the current literature, studies are usually conducted on a set of outdated and incomplete graphs. The majority of the aforementioned graphs are either not easily available or their graph structures are incomplete since they cannot record all the interactions in graph. Moreover, even through all the interactions are recorded, they might not be shareable in a public and timely evolving manner, such as social networks in companies like Meta and Tencent. Thus, for meaningful temporal graph analysis and benchmarks, we need open graph datasets that evolve dynamically and are easily accessible in a timely manner.

To bridge this gap, we propose the concept of Live Graph Lab, which provides live graphs according to blockchain transactions. Specifically, we offer a set of tools for *downloading*, *parsing*, *cleaning*, and *analyzing* blockchain transactions to empower the analyses of transaction graphs. It not only alleviates

37th Conference on Neural Information Processing Systems (NeurIPS 2023) Track on Datasets and Benchmarks.

Table 1: Comparisons among different types of graph datasets.

| Categories | Datasets | Open | Timely Evolving | Complete Structure | Timestamp |
|---|---|---|---|---|---|
| Social Network | ego-Twitter [45] | ✓ | ✗ | ✗ | ✗ |
| Citation Network | DBLP [70] | ✓ | ✗ | ✗ | ✓ |
| The Web | web-Google [44] | ✓ | ✗ | ✗ | ✗ |
| Blockchain | Live Graph Lab | ✓ | ✓ | ✓ | ✓ |

the researchers' burden of accessing massive raw transaction data, but also brings a considerable of opportunities to conduct experiments in the real-world scenario for temporal graph studies. Our introduced live graphs have several unique characters like *open availability*, *dynamic evolution* and *real transactions* due to the inherent characteristics of the decentralized blockchain. Therefore, it is of great importance to investigate the properties of these live graphs to provide new insights for graph algorithms and systems.

Today, as blockchain technology becomes more widespread, the token economy is gradually emerging. Non-fungible tokens (NFTs) have seen tremendous growth with its market capitalization reaching over $40 billion. Notably, one of the digital work named "Everydays: The First 5000 Days" [1] by artist *Beeple* was sold for $69 million, which makes NFTs become the center of attention. This phenomenon also leads to an increasing number of enthusiasts participating in this emerging concept. At the same times, it generates massive, anonymous and real transaction activities, which naturally forms a complex NFT transaction network. Although traditional networks like social networks and citation networks have been extensively studied, they are often outdated due to *their lack of constantly updating*. To enrich the current datasets and overcome their limitations, we instantiate a live graph with NFT transaction network by synchronizing a full Ethereum node, thus it continuously keeps up with the latest Ethereum block and includes all the transaction data. To investigate the characteristics of this live NFT transaction network, we present a temporal graph extracted from a specific time period spanning from 2017 to 2022, which comprises over 4.5 million nodes and 124 million edges. Then, comprehensive analyses are performed, and the results demonstrate that our presented live graph exhibits a variety of characteristics, offering exciting opportunities for the graph community. To summarize, the main contributions and findings are as follows:

- We introduce the concept of live graph lab, which focuses on open, dynamic and real graphs.
- We instantiate a live graph with NFT transaction network and provide a systemic analysis, which demonstrates interesting properties like fast-growing, highly-active, ect.
- Graph machine learning models are investigated in the live graph, and the experimental results indicate that live graphs pose new challenges and opportunities for the graph community.

## 2 Related Datasets

Graph has gained significant attention from both academic and industrial communities. A wide range of benchmark datasets have been proposed to facilitate the research in graph community. Among them, SNAP [43] and Network Data Repository [60] provide diverse types of graphs including social networks, web networks, etc. AMiner [70] offers comprehensive citation networks extracted from DBLP, ACM and Microsoft Academic Graph. Chartalist [63] presents a set of blockchain datasets to enable machine learning model development. Although these datasets are publicly available, they are not constantly updated in a nearly real-time manner. Take social network as an example, the ego-Twitter [45] in SNAP project was released more than 10 years ago. Thus, the graph properties presented in these benchmarks may no longer be suitable in the current context. Meanwhile, their graph structures are usually incomplete due to privacy policy constraints or technical limitations. However, these characters are important for various downstream applications. For instance, if the graph is incomplete or their characteristics have changed significantly, the learning outcomes could be ineffective and even misleading in the graph learning tasks. To enrich the current datasets and overcome their limitations, we propose the concept of Live Graph Lab, which supports various experiments for temporal graph algorithms. We provide a detailed comparison of these datasets in Table 1. Specifically, the proposed live graph lab has the following properties: (1) it is open and

---

[1]https://en.wikipedia.org/wiki/Everydays:_the_First_5000_Days

Table 2: Statistics of the dataset.

| Descriptions | Statistics |
|---|---|
| Start date (mm-dd-yyyy, UTC) | 07-12-2017 13:49 |
| End date (mm-dd-yyyy, UTC) | 08-01-2022 06:50 |
| Number of NFT collections | 97,667 |
| Number of NFT tokens | 77,991,885 |
| Number of account addresses | 4,531,020 |
| Number of transactions | 124,660,813 |

publicly available; (2) it is constantly evolving in a nearly real-time manner; (3) it is complete (i.e, all interactions are fully recorded); (4) it has realistic timestamp. Moreover, previous blockchain datasets mainly focus on fungible token transactions. However, Non-Fungible Tokens (NFTs), which are a vital component of the Ethereum, have been overlooked by existing works. Our work covers this gap by delving into this emerging NFT ecosystem.

# 3 Dataset Details

In this paper, we instantiate a live graph with NFT transaction network in the Ethereum blockchain. We first provide an overview of the blockchain background. Then, we give a comprehensive explanation of the graph construction process. Note that, our methodologies are applicable to other bolckchains like Solana and Polygon, etc.

## 3.1 Background

**Blockchain and Ethereum.** Blockchain, a distributed ledger technology, has attracted continuous attention recent years and is made up of securely linked blocks with cryptography techniques [53], where each block contains information of the previous block (e.g., cryptographic hash). Ethereum is a decentralized, programmable blockchain, which means users can construct various decentralized applications on the blockchain. Ether (ETH) is the native cryptocurrency of Ethereum, and every transaction incurred in the Ethereum needs a specific fee paid in ETH.

**Smart Contract and Non-Fungible Token.** Smart contract is an important feature in the Ethereum blockchain [88], which is a computer program that runs on the Ethereum to automatically execute or control relevant events and actions according to its logic. Smart contracts have largely reduced the requirements for trusted intermediaries, fraud losses and arbitration costs, etc. Non-Fungible Token (NFT) is one of the most successful applications on Ethereum. NFTs are tokens that can be used to represent the ownership of any unique asset such as image, video and audio, etc. Different from fungible items where one dollar is exchangeable for another one dollar, NFTs are not interchangeable with each other, since they all have unique properties and are not divisible.

## 3.2 Raw Data

The statistic information of our dataset is summarized in Table 2. To access the transaction information in Ethereum, Geth, a golang implementation of Ethereum client software, is launched to facilitate the synchronization of the ledger on Ethereum mainnet. When the client synchronizes to the latest block, we extract all the blocks before Aug 1st, 2022 (i.e., from block #0 to block #15,255,104). Then, we parse all the transaction data and log data via toolkit Ethereum ETL[2]. Thanks to the well-defined standards in NFT communities, it is convenient for us to extract the information we need. Specifically, according to the standard of EIP-721, every NFT smart contract must implement the standard interfaces including *transfer*, *approval*, *ownerof* and *balanceOf*, etc. For those smart contract that do not strictly follow the EIP-721 standard, we remove those smart contracts and its relevant transactions from our data. For the remaining data, we filter out all the *transfer* events triggered by its smart contracts to extract the NFT transactions. This is because the ownership change of any NFT will emit a *transfer* event identified by the event topic Keccack256 hash *0xddf...3ef*. In the NFT *transfer* event, it contains four key contents including the Keccack256 hash, the sender's address, the receiver's address, and the transferred NFT token ID. The time when the transaction occurs can be

---

[2]https://ethereum-etl.readthedocs.io/en/latest/

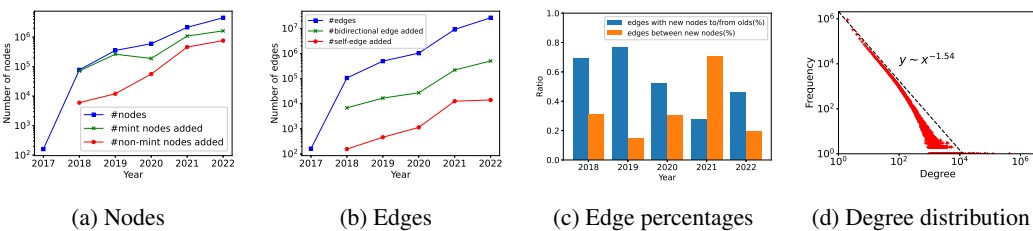

| (a) Nodes | (b) Edges | (c) Edge percentages | (d) Degree distribution |

Figure 1: Evolution of nodes and edges.

retrieved from the block that the event was found. Once we have obtained all of these key information, we can know when and which NFTs are transferred among different wallet addresses and the prices they are sold.

By parsing the transfer events, we find out that there are 97,667 NFT collections with 77,991,885 NFT tokens, where each collection contains different number of NFT tokens (e.g., varying from thousands to millions). Meanwhile, 124,660,813 transactions are extracted, and among them 4,531,020 users (i.e., we regard each wallet address as a user) participate in the transactions. It is worthy noting that there are over 100,000 NFT collections according to Etherscan[3]. However, as we have mentioned, some of them are not standard NFT tokens (i.e., they do not strictly follow the EIP-721 standard), and we remove them from our data. Thus, our method almost extracts all the NFT collections.

### 3.3 Graph Construction

We investigate the structure and dynamics of all these transactions by constructing a directed temporal graph $\mathcal{G} = (\mathcal{N}, \mathcal{E}, \mathcal{T})$, where $\mathcal{N}$ and $\mathcal{E}$ denote the node set and edge set, respectively. That's to say, we only count once for those addresses that have repetitive interactions. $\mathcal{T}$ is a set of timestamps when the interactions happen. We use $t(e)$ to denote the timestamp when edge $e$ is formed, and $t(u)$ represents when the node $u$ is added into the graph. Thus, $a_t(u) = t - t(u)$ reflects node $u$'s age at time $t$. For any given time $t$, graph $\mathcal{G}_t$ consists of all the nodes as well as edges until time $t$. Note that, since we have accurate timestamps for the arrival of each node and edge, our investigation of graph dynamics is at a much finer granularity compared with the majority of existing studies [5, 41].

## 4 Observations and Analyses

To have a better understanding of the instantiated live graph, we start the analyses from the following perspectives: (1) Structural properties, which investigate how its nodes and edges change as time goes on; (2) Dynamic behaviors, which are graph specific properties such as how hub-nodes and bi-directional edges are formed. More comprehensive analyses are given in the Appendix C and D.

### 4.1 Structural Properties

To fully understand how active is the NFT transaction network, we measure the evolution of nodes and edges over the time. Specifically, we set the time granularity to a yearly yardstick. Since our data started from the July of 2017 and ended at the August of 2022, the statistical information for 2017 and 2022 only has a half year's data. Figure 1a and 1b show the annual growth of nodes and edges in a log-scale. As observed, there is a rapid increase in the number of nodes and edges, which become 28K and 165K times larger within the six years. It means the NFT transaction network is highly active and growing at a fast speed.

To further figure out what leads to the growth, we analyse the newly added nodes in each pair of consecutive years. New nodes could join the network through different ways (i.e., mint, buy or airdrop NFTs). Among them, mint is the most common and easy way to obtain NFTs, where only a small number of gas fee is needed to pay. Figure 1a presents the trend of newly added mint nodes and non-mint nodes. We notice that the number of mint nodes added into the network is approximately the same magnitude of the total nodes at that time. Therefore, mint NFTs dominates the growth of

---

[3]https://etherscan.io/tokens-nft

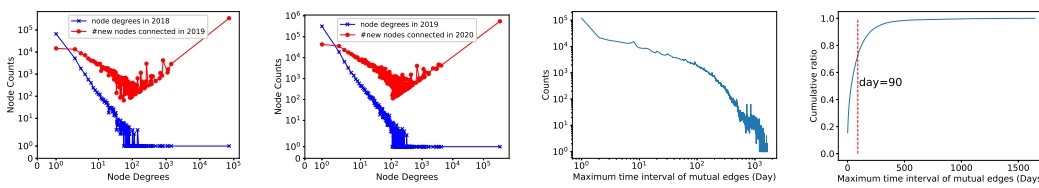

(a) Hub nodes in year 2018  (b) Hub nodes in year 2019  (c) Mutual edges in days  (d) Ratio of mutual edges

Figure 2: Correlation between hub nodes, new connections and mutual edges.

nodes in the network at present, meanwhile the number of non-mint nodes are also increasing at a high velocity. These two phenomena result in the expansion of nodes.

Figure 1b also shows the trend of added bidirectional edges and self-edges. We can see that the number of bidirectional edges only account for a small volume. Unlike social networks where the edges are highly mutual, the NFT market is an anonymous ecosystem and the probability of mutually interacting with each other is relatively low. These bidirectional edges might happen in the scenarios of swap or wash trading activities [73]. We also notice that there exist a very few self-loops in the network, which means these addresses transact with themselves. This abnormal phenomenon might be caused by a mistake input for the received address or these transactions are executed for testing. Furthermore, to characterize how active are these addresses, we compute the percentage of edges in which new nodes have links from or to old nodes, and the percentage of edges which only contain links between new nodes. Here, we refer to new nodes as the nodes created in the current year, and old nodes indicate the nodes that have existed in the previous years. Figure 1c demonstrates that more than 50% percent of newly added edges are constituted by the connections between new nodes and old nodes, except for the year of 2021. This indicates that most of the addresses remain active as the NFT ecosystem become mature. It is also very interesting to note that the newly created addresses are highly active in 2021, and at the same time, the whole NFT market capitalization reaches over $40 billion USD dollars in this year.

For completeness, we also show the degree distribution of the NFT transaction network in Figure 1d with log-log scale. As expected, it follows a power-law distribution. We observe that some nodes have the degree of 1, which indicates they did not conduct any other transactions after the NFTs were minted or transferred. There also exist several high degree hubs that are extremely active and interacts with more than thousands of different addresses. Among them, a special Null address (i.e., *0x000...000*)[4] has the highest degree, since every NFT mint activity will create a link from the Null address to the mint address.

## 4.2 Dynamic Behaviors

**Evolution of Hub Nodes.** The node degree shows a heavily long-tailed distribution (i.e., Figure 1d), and the assortativity raises year by year (i.e., Figure 4a in Appendix C). Both these two measurements are highly relevant to the evolving of hub nodes in the transaction network. Figure 2a and 2b illustrate the correlations between node degrees and its number of new connections in two consecutive years. We use blue color to present the node degree distribution in the previous year, and then red color indicates the number of connections from new nodes in the current year. As expected, we can find that if a node had high degree in the previous year, it would have high probability to get more new node connections in the current year. Moreover, this observation is also validated by measuring their Pearson Correlation Coefficients. We evaluate the correlation coefficients between node degrees and its number of new node connections (in year 2018-2019 and 2019-2020), which are 0.2266 and 0.5476, respectively. The results reveal that these two factors are positively relevant. Thus, we can draw the conclusion that the NFT transaction network follows preferential attachment growth model [54, 32], i.e., "the rich gets richer".

Next, we analyze the distribution of the hub nodes. Specifically, we note that half of the top-100 largest hub nodes are smart contract addresses. This is understandable, since smart contracts play an important role in providing various services (e.g., NFT fractionalization and staking) to other users

---

[4]0x0000000000000000000000000000000000000000

in the network. Thus, they tend to have high degrees once they are frequently used by others. The transactions with such smart contracts result in the increase of assortativity. Furthermore, there also exist some hub nodes that are not smart contracts. For instance, we observe that address *0x8a0...700*[5] is the fifth largest hub nodes with the degree of 69,458. After checking its transactions manually, we find that all its transactions are about HyperNFT tokens, and it holds more than 180,000 such kind of tokens. Since it is strange, we then check its transactions in details and discover that this address creates a transaction every a few minutes, where the ID of the transacted NFT token is in a continuous and increase order. According to these clues, it is very certain that this address is a bot, which makes frequent transactions to pretend it's very prosperous to attract more users to join in. The hub nodes are responsible for the spreading of information in the network, thus it is necessary to investigate the properties of hub nodes.

**Mutual Interaction Edges.** In the previous study, we find that the reciprocity of the transaction network is relatively low, which is around 0.1 in the end (i.e., Figure 4c in Appendix C). This is quite different from the property in the social networks, where the reciprocity is very high and the value is above 0.7 [39]. One possible reason is that people are prone to mutually following and interacting with their friends in the social networks, since it does not cost anything. However, it becomes different in the NFT transaction network, because we do not know who we are actually interacting with and every action costs in the blockchain. To uncover the reciprocity's characteristics, we focus on the mutual interaction edges. Specifically, we are interested in if two nodes are mutually interacting with each other, i.e., reciprocal edges $\langle u, v \rangle_t$ and $\langle v, u \rangle_{\hat{t}}$, what their maximum time interval distribution $|t - \hat{t}|$ looks like, i.e., the maximum delay in days of the reciprocity. Figure 2c shows that mutual interaction edges forming within one day account for the highest proportion, and the maximum time interval can reach to more than 1,000 days. According to Figure 2d, we can observe that about 15% of the reciprocal edges are formed almost simultaneously among those bi-directional edges, and more than 70% of them are formed within 90 days. From these observations, we can conclude that the NFT transaction network can not be regarded as an undirected network due to its low reciprocity value, and the simultaneously formed bi-directional edges can be a good indicator to judge the abnormal activities. For instance, it may be caused by the token transfers among the accounts controlled by a same person.

To further investigate this phenomenon, we want to know how many address pairs are suspicious in these bi-directional links. We first conduct some statistics for all NFT transaction addresses, which involves the following two factors: the number of transactions (including from transactions and to transactions) and the number of distinctly interacted addresses. Then, if one address has very limited transactions and it only interacts with a specific address, it is highly possible that these two addresses are of same person's wallets. This is because this address is actually not active, but it responses promptly in the bi-directional links. Similarly, another situation is although one address has a lot of transactions, it interacts with one specific address frequently and the specific address accounts for a large ratio of the total transactions. They may also belong to same person's wallets. Based on these two observations, we define the following two rules to identify the suspicious address pairs: 1) one of them makes less than 5 transactions or 2) the transaction ratios between them is larger than 0.8. If they satisfy at least one of the rules, we then believe these two addresses are likely to be same person's wallets. According to these rules, we discover that 33.72% of those simultaneously formed bi-directional links have high probability of being suspicious. This observation provides a new perspective for detecting anomaly transactions.

## 5 Downstream Applications

The above analyses provide a general overview of this live graph's properties. Our results demonstrate that the transaction network is highly active and evolving at a fast speed. These properties put forward new challenges for various downstream tasks. Next, we investigate three widely studied tasks.

### 5.1 Temporal Link Prediction

Link prediction lies at the core of graph analysis and mining. The link prediction's goal is to predict whether a pair of node would form a link, which has been extensively used in diverse real-world scenarios like recommendation [79, 7] and knowledge graph completion [55], just to name a few. In

---

[5]0x8a01fa5a77311bbcf29e293d8ecb48707cfdb700

this section, we focus on temporal link prediction, i.e., we try to forecast future interactions based on the historical transactions. Specifically, we investigate the snapshot-based representation for temporal link prediction via graph neural networks [36, 25]. Since each edge $e$ has a timestamp $\tau_e$, we utilize a sequence of graph to depict the temporal transaction network $G = \{\mathcal{G}_t\}_{t=1}^T$, where each snapshot is a static graph $\mathcal{G}_t = (\mathcal{N}_t, \mathcal{E}_t, \mathcal{T}_t)$ with $\mathcal{E}_t = \{e \in \mathcal{E} | \tau_e \leq t\}$. Through modeling the sequential information of different graph snapshots, we could leverage the information available up to time $t$ to forecast possible edges at time $t + 1$. Although various approaches have been proposed for temporal link prediction, it is unclear whether they can achieve satisfied performance in this highly active and large-scale temporal transaction network.

**Graph Neural Networks.** GNNs have gained great success in numerous learning tasks on graphs including node classification [36, 25], graph classification [84, 83] and link prediction [82, 75]. The objective of GNN is to acquire effective node representations through iteratively aggregating messages from its local neighborhood. Specifically, the $l$-th layer of a GNN model can be formulated as follows:

$$\mathbf{h}_v^l = \text{AGG}^l \left( \left\{ \text{MSG}^l(\mathbf{h}_u^{l-1}, \mathbf{h}_v^{l-1}) | u \in \mathcal{N}(v) \right\}, \mathbf{h}_v^{l-1} \right)$$

where $\mathbf{h}_v^l$ is the node representation for node $v$ at layer $l$ and $\mathcal{N}(v)$ represents node $v$'s neighborhood. $\text{MSG}(\cdot)$ indicates the message-passing function, which propagates information from its neighbors. $\text{AGG}(\cdot)$ is the aggregation function, which updates its representation with node's neighborhood representations. For a $L$ layers GNN, it aggregates information from the $L$-hop neighborhood. Different GNN architectures can have different message-passing and aggregation functions. The original GNN model is designed for static graphs, thus it cannot capture the underlying temporal information within the graph. To incorporate the evolving property, existing works utilize RNNs [28, 14] to aggregate information from different graph snapshots. Hence, the dynamic GNN models have much more parameters compared with the vanilla GNNs, which is more difficult to scale to large graphs due to the back propagation through time constraints.

**Models.** We compare a number of recent state-of-the-art dynamic GNN models. (1) Dyngraph2vec [21] captures the temporal transitions with a deep architecture with dense and recurrent layers. (2) TGCN [87] integrates GCN with GRU to learn complex spatial and temporal dependencies. (3) EvolveGCN [56] use a RNN to adapt the graph convolutional network parameters. (4) GCRN [62] generalize TGCN with either GRU or LSTM. Moreover, it uses ChebNet [15] to encode the graph structure information, and separate GNNs are applied to compute different gates of RNNs. (5) DynGEM [22] utilizes deep autoencoder to perform graph embedding, then local and global constraints are employed to keep node representations being stable over time. (6) Roland [81] is an efficient learning framework designed for temporal GNNs, which updates its parameters through a combination of incremental training and meta-learning strategies [19, 20].

**Settings.** For dataset, we remove all the transactions associated with the Null address, which results in 3.13 million nodes and 23.13 million edges in the directed graph. We set node feature as 1 for all the nodes and utilize area under the curve (AUC) as well as mean reciprocal rank (MRR) as evaluation metrics. For every node $u$ connected by a positive edge $(u, v)$ at time $t$, we randomly select 100 negative edges originating from node $u$. Subsequently, we determine the rank of the score for edge $(u, v)$ among all the sampled negative edges. AUC characterizes the probability of ranking positive node more highly than negative nodes. MRR denotes the average of reciprocal ranks computed across all nodes. Following the settings in Roland [81], we utilize two different train-test splits: fixed-split and live-update. Among them, *the fixed-split setting assesses the models using all the edges from the last $20\%$ graph snapshots*. Although it is widely used in the existing works, the fixed-split might produce misleading results based on edges merely from the last few graph snapshots, since the graph structure could constantly evolve in the real world scenario. *To eliminate this bias, we also utilize live-update split to test the models, which evaluates their performance over all the available graph snapshots*. $10\%$ of edges are used to determine the early-stop condition in this setting.

**Results.** We present the link prediction results in Table 3. Specifically, we use three different time granularities (i.e, days, weeks and months) to construct the graph snapshot sequences, which results in 1,657 graph snapshots, 253 graph snapshots and 60 graph snapshots, respectively. All the models' performance is evaluated under two different settings, i.e., fixed-split and live-update. We notice that as the time granularity becomes coarser, the models' performance drops. This is because the NFT transaction network is highly active, which is about 68 million transaction volume per day on average in 2021. Thus, it will be more difficult to predict all the possible links in a long time period. Meanwhile, we can observe very high AUC scores in a daily time granularity, which indicates the

Table 3: Temporal link prediction performance in fixed split and live-update settings. We repeat experiments with three different seeds to report the mean as well as standard deviation of AUC and MRR. We also present the results under different time snapshot granularities, e.g., days, weeks and months. OOM means out-of-memory.

| Models | Fixed Split | | | | | |
| | Snapshot Days | | Snapshot Weeks | | Snapshot Months | |
| | AUC | MRR | AUC | MRR | AUC | MRR |
|---|---|---|---|---|---|---|
| Dyngraph2vec | OOM | OOM | OOM | OOM | OOM | OOM |
| TGCN | 53.64±1.60 | 14.24±2.60 | 61.55±6.24 | 36.16±6.60 | 74.87±4.99 | 45.97±4.46 |
| EvolveGCN | OOM | OOM | OOM | OOM | OOM | OOM |
| GCRN-GRU | 95.86±0.03 | **71.48±0.49** | 93.14±0.18 | **68.44±0.05** | **86.74±0.80** | 58.23±1.23 |
| GCRN-LSTM | 94.12±0.92 | 68.51±2.36 | 92.90±0.43 | 67.66±0.31 | 86.44±0.92 | 58.71±0.84 |
| DynGEM | OOM | OOM | OOM | OOM | OOM | OOM |
| Roland-MA | **95.93±0.15** | 66.34±0.23 | **93.53±0.13** | 65.06±0.43 | 86.23±0.85 | 54.93±1.42 |
| Roland-MLP | 65.46±6.10 | 43.76±5.94 | 73.34±7.87 | 42.04±16.8 | 85.88±2.22 | 57.58±4.12 |
| Roland-GRU | 73.33±11.5 | 49.45±8.91 | 91.48±1.67 | 66.28±1.86 | 86.59±0.88 | **59.37±1.54** |
| | Live Update | | | | | |
| Dyngraph2vec | OOM | OOM | OOM | OOM | OOM | OOM |
| TGCN | 58.22±5.76 | 17.77±10.7 | 59.94±8.44 | 22.67±19.3 | 75.01±2.66 | 43.16±0.75 |
| EvolveGCN | OOM | OOM | OOM | OOM | OOM | OOM |
| GCRN-GRU | 80.95±1.92 | 39.13±0.39 | 85.34±0.26 | 46.08±1.43 | 81.40±0.34 | 43.68±0.39 |
| GCRN-LSTM | 79.12±2.14 | 37.83±1.16 | 84.73±0.34 | 42.89±3.34 | 81.24±0.80 | 41.47±3.00 |
| DynGEM | OOM | OOM | OOM | OOM | OOM | OOM |
| Roland-MA | **90.47±0.66** | **49.79±0.95** | **88.74±0.37** | **50.77±1.13** | 83.93±0.95 | 47.11±1.16 |
| Roland-MLP | 56.32±8.06 | 22.16±15.4 | 70.88±10.5 | 40.91±10.8 | 79.38±4.47 | 46.51±2.61 |
| Roland-GRU | 60.04±5.08 | 27.29±6.26 | 75.93±19.2 | 48.66±13.7 | **84.36±0.46** | **50.75±0.83** |

models have a better capability of distinguishing the positive and the negative edges in this short time scenarios. Also, the performance in live-update setting is a little lower than the fixed-split setting. This is caused by the pattern shifts in different graph snapshots, which implies the patterns indeed evolve over the time. Therefore, we can conclude that it is necessary to model the NFT transaction network in a finer time granularity, and at the meantime this will result in longer graph snapshot sequence, which puts forward more challenges to model the temporal information.

## 5.2 Temporal Node Classification

Node classification plays a crucial role in understanding the attributes, behaviors, and relationships of nodes in temporal graphs, offering valuable insights in diverse applications. By predicting the label of nodes at a particular time $t$, we gain a temporal perspective that deepens our understanding of how nodes evolve over time and their dynamic characteristics. Specifically, we focus on categorizing nodes according to their transaction behaviors. They can be generally classified into five distinct classes: daily traders, weekly traders, monthly traders, yearly traders and the remaining traders. We first filter out nodes that only have one transaction. Then, each node' maximum transaction interval is calculated. If the maximum interval is within one day, we call it daily trader. Likewise, if the maximum interval is within one week and larger than one day, we call it weekly trader, and so forth. This process results in a large-scale directed graph with about 1.80 million nodes and 21.83 million edges. Following the setting of EvoloveGCN [56], we use the first 80% graph snapshots as training set, the following 10% graph snapshots as validation set and the last 10% graph snapshots as test set. Similar to temporal link prediction task, we use the same set of GNN models and add a multi-class classification layer. Furthermore, nodes' degrees are encoded as features. The number of transactions between two nodes and their latest interaction timestamp are transformed as edge features. Two commonly used metrics (i.e., accuracy and recall) are employed to assess the model's performance.

**Results.** The node classification results are illustrated in Table 4 and key observations are as follows. Three granularities, i.e., days, weeks and months, are utilized to generate the temporal graph snapshot sequences. In accordance with the temporal link prediction task, we can draw similar conclusions. Roland could consistently outperform other baselines, and its variant Roland-MA (moving-average) stands out due to its parameter-free design and remarkable capability to capture evolving patterns. On the other hand, TGCN fails to achieve satisfactory performance because of the gradient vanishing problem in longer sequences. GCRN-GRU and Roland-GRU address this issue by utilizing separate

Table 4: Node classification performance in fixed split setting. We repeat experiments with three different seeds to report the mean as well as standard deviation of Accuracy and Recall.

| Models | Snapshot Days | | Snapshot Weeks | | Snapshot Months | |
|---|---|---|---|---|---|---|
| | Accuracy | Recall | Accuracy | Recall | Accuracy | Recall |
| Dyngraph2vec | OOM | OOM | OOM | OOM | OOM | OOM |
| TGCN | 18.48±2.66 | 31.15±3.16 | 43.97±4.57 | 32.99±2.34 | 47.45±3.49 | **32.53±2.95** |
| EvolveGCN | OOM | OOM | OOM | OOM | OOM | OOM |
| GCRN-GRU | 41.06±3.30 | 34.75±2.93 | 46.78±0.72 | **34.79±0.42** | 47.42±2.16 | 28.97±3.22 |
| GCRN-LSTM | 46.14±3.29 | **35.19±3.61** | 48.04±2.37 | 31.58±1.75 | 49.32±2.01 | 35.39±1.49 |
| DynGEM | OOM | OOM | OOM | OOM | OOM | OOM |
| Roland-MA | **51.02±2.01** | 28.77±3.23 | **50.39±0.45** | 26.33±3.95 | 47.96±2.69 | 22.33±3.07 |
| Roland-MLP | 48.46±3.18 | 30.62±3.94 | 47.59±3.39 | 31.67±3.62 | 45.74±4.75 | 35.04±3.47 |
| Roland-GRU | 49.88±2.15 | 33.38±3.91 | 46.63±3.07 | 33.85±1.48 | **50.04±0.37** | 32.17±2.72 |

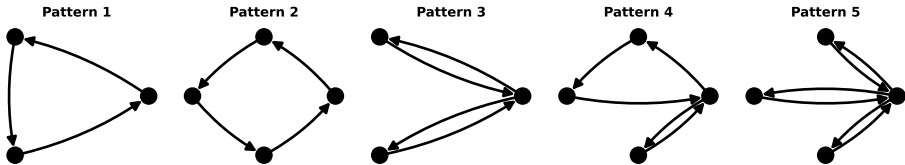

Figure 3: Five most common wash trading patterns.

GNNs or leveraging previous and current snapshots. Furthermore, the scalability of several baselines is limited and they experience out-of-memory (OOM) issues. Among the three time granularities, the month granularity is generally the most difficult scenarios. This is because with a month granularity, the time period between snapshots is longer compared with the day and week granularities. As a result, it becomes more challenging to accurately predict the temporal patterns and changes in node behaviors. The longer time gap between snapshots increases the complexity of modeling the evolving dynamics of the network and introduces more uncertainty, leading to decreased performance in terms of classification accuracy and recall. Therefore, we can conclude that given the highly active NFT transaction network, it is essential for the model to use a finer time granularity. Nonetheless, this choice results in longer graph snapshot sequences, presenting additional difficulties in accurately capturing the temporal information.

## 5.3 Continuous Subgraph Matching

Continuous Subgraph Matching (CSM) plays a vital role in numerous real-time graph applications. The goal of CSM is to identify and report the occurrences of a given query graph $Q$ within a temporal graph stream $G$. It can be utilized in various scenarios. For example, when setting the query graph $Q$ as wash trading patterns in e-commerce, CSM could identify the anomaly transaction patterns in the graph via exactly matching [58]. Similarly, representing query graph as rumor patterns in social network could help to detect and prevent the spread of rumors [74]. In this subsection, we set the query graphs as the frequent wash trading patterns in the NFT transactions. According to the definition in [73, 47], wash trading is an activity where the seller is on the both sides of the trade. The goal of wash trading is to influence the price or create the illusion that the item is very popular, which produces artificial activities in the marketplace. Wash trading has been prohibited by many countries. However, due to the anonymous nature of the blockchain, wash trading has become a severe issue in the NFT market[6], which accounts for a large volume in the whole NFT transactions. For instance, the CryptoPunk's $9,998$-th NFT was traded between two wallets for 124,457 ETH (about $534 million USD), in which the buyer paid to the seller, then the seller transferred the money back to the buyer. Thus, it is important to detect the wash trading transactions to uncover the anomaly behaviors and reduce the risks in the market. We resort to CSM to identify the wash trading transactions. Five most common wash trading patterns are shown in Figure 3. As can be seen, all of them contain at least one cycle. Here is a toy example that exemplifies Pattern 1: Address A (*0x744...282*) initiated the sale of Azuki token 1,215 to Address B (*0xd39...263*). Subsequently, Address B sold the token to Address

---

[6]https://cryptopotato.com/over-33-of-nft-volume-is-wash-trading-bitscrunch-ceo-interview/

Table 5: Comparison of different frameworks on query time.

| Query Patterns | | Model Query Time (ms) | | | |
|---|---|---|---|---|---|
| Queries | Counts | SymBi | Graphflow | TurboFlux | RapidFlow |
| *p1* | 19,338 | $1.22\times10^4$ | $1.11\times10^4$ | $1.11\times10^4$ | $\mathbf{5.93\times10^2}$ |
| *p2* | 2,243,232 | $1.19\times10^4$ | $1.44\times10^4$ | $1.51\times10^4$ | $\mathbf{6.13\times10^2}$ |
| *p3* | 3,012,738 | $1.11\times10^4$ | $1.13\times10^4$ | $1.08\times10^4$ | $\mathbf{5.83\times10^2}$ |
| *p4* | 9,472,960 | $1.24\times10^4$ | $1.43\times10^4$ | $6.06\times10^4$ | $\mathbf{6.45\times10^2}$ |
| *p5* | 3,154,355,868 | $1.34\times10^5$ | $4.24\times10^5$ | $7.72\times10^4$ | $\mathbf{5.84\times10^2}$ |

C (*0xeaa...c0f*), and eventually, Address C sold it back to Address A. These transactions happened within half an hour, and interestingly, the token's price surged from 8.98 ETH to 11.99 ETH. Such activity can raise suspicions of wash trading. We will use them as query graphs in the experiments to simulate the wash trading detection procedure. More details are given in Appendix E.

**Results.** Table 5 shows the key results of continuous subgraph matching. As we can see, RapidFlow [68] is the most efficient algorithm, which demonstrates about 18-724x speedups compared with the remaining frameworks. This is because the query reduction technique can significantly expedite the query procedure through an optimized matching order. We also observe that no algorithm can dominate others in different query patterns except RapiadFlow. Among them, SymBi [50] and Graphflow [33] perform quite worst on pattern 5, which need 10x more time to produce the results. SJ-Tree [13] and IEDyn [30, 31] cannot output the results within the time limit. The reason is that SJ-Tree encounters memory exhaustion in the majority of cases, and IEDyn's index update bears too much overhead because of the maintenance for constant-delay enumeration. Furthermore, the number of matched subgraphs increase from pattern 1 to pattern 5, however the query time does not have too much difference in almost all the frameworks. We can conclude that these four frameworks are applicable to large-scale graphs with dense structures. They can effectively serve as a bridge to generate ground truth for continuous subgraph matching, providing valuable support for the training of deep graph learning models.

## 6 Conclusion

In this paper, we propose the concept of Live Graph Lab, which includes blockchain based temporal graphs that are openly accessible, fully recorded, and dynamically evolving over time. Specifically, we instantiate a live graph using the NFT transaction network and investigate its dynamic properties in a temporal graph analysis perspective. Our findings reveal both similar and distinct characteristics when compared to traditional networks like social networks, citation networks, and the Web. Through comprehensive experiments on the live graph, we uncover numerous insightful discoveries. The proposed live graph overcomes the limitations of existing datasets and enhances the diversity in graph research. We believe that the live graph lab will become an indispensable resource for the graph community and open up new opportunities. There are also various other potential use cases, such as identifying whether an account holds a specific type of tokens or predicting the range of tokens that the account possesses, etc.

## 7 Border Impact and Limitation

The Live Graph Lab can facilitate researchers from graph community by offering comprehensive blockchain based graphs via an easily accessible manner. Insights from this research can be directly applied to improve the design, security, and user experience of NFT platforms, leading to sustainable growth of the ecosystem. Meanwhile, as the live graph constantly records all the NFT transactions in the Ethereum blockchain, the possibility of encountering malicious activities could become a concern, such as bot transactions or wash trading, etc. It might also cause potential negative societal impacts. Since the dataset consists of complete transactions associated with the wallet addresses, it could enable the tracking of each wallet's behaviors, habits, and financial activities. This kind of tracking could be exploited for targeted advertising, manipulation, or surveillance. The dataset could also make it possible for malicious actors to analyze the transaction patterns and manipulate the NFT market, which could lead to unfair practices, price manipulation, and market instability.

## Acknowledgments and Disclosure of Funding

This research is supported by the National Research Foundation, Singapore under its Industry Alignment Fund – Pre-positioning (IAF-PP) Funding Initiative. Any opinions, findings and conclusions or recommendations expressed in this material are those of the author(s) and do not reflect the views of National Research Foundation, Singapore. The authors would like to thank reviewers for their helpful comments. The authors would also like to thank Zhengtao Jiang for the development of website.

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

# A Detailed Background

In this section, we explain the terminologies related to Ethereum blockchain, transactions and NFTs, etc.

## A.1 Blockchain and Ethereum

Blockchain, a distributed ledger technology, has drawn continuous attention recent years. Blockchains are made up of securely linked blocks with cryptography techniques [53], where each block contains information of the previous block (e.g., cryptographic hash). Then, consensus algorithms or protocols are applied to validate transactions and keep them being consistent. By this way, the blockchain transactions are immutable, traceable and publicly available. Ethereum is a decentralized, programmable blockchain, which means users can construct various decentralized applications on the blockchain. Ether (ETH) is the native cryptocurrency of Ethereum, and every transaction incurred in the Ethereum needs a specific fee paid in ETH. According to the market capitalization, ETH is the second-largest cryptocurrency behind Bitcoin.

## A.2 Account and Transaction

Ethereum account is the key to access and explore the Ethereum ecosystem. Every Ethereum account is associated with a unique address, akin to an email address for an inbox. The address can be used to receive or send funds to the corresponding account. Accounts in the Ethereum can be classified into two types: (1) Externally-owned account (EOA) and (2) Contract account. All the accounts are denoted as a 64 character hex string.

Transactions are messages sent from one account to another account. One of the simplest transaction is transferring ETH from one account to another, which will change the state of the Ethereum Virtual Machine (EVM) and need to be broadcast to the whole Ethereum network. Each transaction requires amount of fees to pay for the computation. The key information in a transaction includes the receive address, the sender address, value, data, gas limit, and the max fee per gas, etc. There are three categories of transactions within the Ethereum: 1) regular transactions, which indicates the transactions between two externally-owned accounts; 2) contract deployment transactions, which are special transactions without receive addresses; 3) execution of a contract, whose receive address is the smart contract address. When the transaction is submitted, its life cycle can be simplified into the following three steps: 1) an externally-owned account sends a transaction and generates a transaction hash; 2) the transaction is broadcast across the network; 3) a validator verifies the transaction and includes it into a block. Once the transaction is successfully executed, it can never be altered.

## A.3 Smart Contract and Non-Fungible Token

Smart contract is an important feature in the Ethereum blockchain [88], which is a computer program that runs on the Ethereum to automatically execute or control relevant events and actions according to its logic. Smart contracts have largely reduced the requirements for trusted intermediaries, fraud losses and arbitration costs, etc. As we have mentioned before, smart contracts also belong to a type of Ethereum account, and it can interact with user accounts. Moreover, anyone can program a smart contract and deploy it to Ethereum, as long as the code is complied successfully and can be executed by the EVM. Smart contracts are the fundamental building blocks for various applications like decentralized finance (DeFi) and game finance (GameFi).

Non-Fungible Token (NFT) is one of the most successful applications on Ethereum. NFTs are tokens that can be used to represent the ownership of any unique asset such as image, video and audio. Different from fungible items where one dollar is exchangeable for another one dollar, NFTs are not interchangeable with each other, since they all have unique properties and are not divisible. Smart contracts manage the ownership and the transferability of NFTs. Specifically, when NFTs are minted or transferred, it triggers the code stored in the smart contract, and then the relevant actions are executed. Each NFT token will have an owner after mint, and this information can be easily verified in Ethereum. The NFTs can be bought and sold on any NFT market like OpenSea, LooksRare or X2Y2. More recently, the NFT and DeFi have been combined to form a number of interesting applications including NFT-backed loans, fractional ownership and certificates of authenticity.

# B Related Work

In this section, we present the related work on graph analysis.

**Analyses of Social Networks, Citation Networks and the Internet.** There exist lots of prior works focusing on analysing social networks and citation networks like Flickr, Yahoo! 360 and LiveJournal [5, 38, 39, 41]. These studies investigate the graph properties including density, degree distribution and clustering coefficient, etc. Among them, [17] observed that the node degrees followed a power-law distribution in most real-world networks. [9] studied the graphs from the perspective of connectivity, where large strongly connected components (i.e., SCC) widely existed in the graphs. [23] classified social network's links into strong ties and weak ties, with strong ties indicating tighter clustering. [77, 37] explained the social networks' small-world phenomenon. [41] showed that the citation graphs demonstrated denser densities and decreasing diameters as time goes by. Then, a forest-fire graph generation model was proposed to simulate these phenomena. [64] proposed a jellyfish model to describe the topology of the Internet, which abstracted the structure in a human understandable way.

To summarize, the major findings of existing works are as follows: 1) power law degree distribution, where some nodes exhibit significantly large degrees; 2) preferential attachment growth model, where the likelihood of a new node establishing a connection with an existing node is directly tied to the degree of the existing node; 3) density of the graphs follows a rapid decline, and then becomes steady. For surveys of graph analysis, interested readers can refer to [27, 76]. Although the graph analyses are extensively studied in those networks, it is not clear whether these findings are still valid in this emerging NFT transaction network.

**Graph Analyses of Cryptocurrency Transaction Networks.** Due to the decentralized nature of blockchain, this makes it possible for everyone to access all the transaction information. Several recent works have studied the properties of Bitcoin and other cryptocurrency transaction networks. For instance, [26, 59, 48] studied the user behaviors in Bitcoin transaction network. [66] classified and visualized the information extracted from the Bitcoin network. [80] and [1] forecast the price of BTC via modeling the local topological structure of the Bitcoin graph. Apart from Bitcoin, other cryptocurrencies like Zcash [34], EOS [29] and Monero [51] had also been conducted similar analyses. [8] analyzed the transaction linkability in Zcash, which revealed the underlying privacy concerns. [16, 2] discussed the key factors that impact the scalability of various blockchain systems. [24] identified arbitrage behaviors among multiple cryptocurrency exchange markets (e.g., Kraken, Coinbase and Gemini) through weighted cycle detection. However, the majority of these works are performed on analyzing static graphs, whereas graphs are usually evolving over time in the real-world scenarios. Moreover, the aforementioned studies that analyze Bitcoin and other cryptocurrencies only involve transactions related to value or token transfers. Different from Bitcoin, recent popular blockchains like Ethereum and Solana support deploying smart contracts to provide diverse services, where human controlled accounts and program controlled agents coexist in the network, making the transaction network even more complicated. It is of great interest to us to investigate this type of transaction network.

**Analyses of the Ethereum Blockchain.** Given the possibility to access comprehensive information within the blockchain transaction network, some recent efforts follow the pioneer studies on social networks, citation networks and the Internet [39, 41, 64] to analyze the static Ethereum transaction network. Particularly, [40] measured four interaction networks to give new insights on the Ethereum graph properties. [18, 11] characterized major activities including money transfer and contract creation on Ethereum via graph analysis. [46] learned from Ethereum graph to perform price anomaly prediction. [86, 6] investigated the evolutionary dynamics of Ethereum activities through the lens of temporal graphs.

Instead of studying all the transactions in the Ethereum, [65] only considered transactions relevant to ERC20 tokens which were fungible tokens circulated in the Ethereum, and the results showed that they presented obvious social signals in the trading network. [12] performed a systematic analyses on the whole ERC20 token activities. [72] studied massive individual token networks from a graph analysis perspective, and found that they were largely dominated by a single hub and spoke pattern. Since tokens could be heavily influenced by various events like mint, burnt, transfer or staking, these aforementioned works only provide a general intuition about the structure of token distributions, the flow and spread of assets on the blockchain.

Similar graph analysis approaches have also been applied to Non-fungible tokens (NFTs), where NFT tokens are unique and not comparable with each other. For example, [78] used the Louvain algorithm [71] to extract the community based structures from the networks, which characterized the relationships between buyers and sellers. [52] showed that the NFT's sale history and visual appearance were two good indicators to predict its price. [10] analyzed several popular NFT projects, and concluded that the structure of NFT networks was qualitatively similar to social networks. [73] quantified suspicious wash trading behaviors in NFT market via closed cycle detection in the network. Although NFTs play a crucial role in the Ethereum ecosystem, none of the aforementioned studies explore the temporal properties of the NFT transaction network from the temporal graph point of view.

## C  Basic Structure Properties

The following four properties are discussed in this section:

**Assortativity** characterizes the tendencies of nodes getting attached to similar nodes through a specific metric. Following [4], we calculate the degree assortativity as follows:

$$\alpha = \frac{\frac{1}{|\mathcal{E}|} \sum_{(i,j) \in \mathcal{E}} k_i k_j - [\frac{1}{|\mathcal{E}|} \sum_{(i,j) \in \mathcal{E}} \frac{1}{2}(k_i + k_j)]^2}{\frac{1}{|\mathcal{E}|} \sum_{(i,j) \in \mathcal{E}} \frac{1}{2}[k_i^2 + k_j^2] - [\frac{1}{|\mathcal{E}|} \sum_{(i,j) \in \mathcal{E}} \frac{1}{2}(k_i + k_j)]^2} \tag{1}$$

where $k_i$ and $k_j$ represent the degrees at the ends of edge $(i, j) \in \mathcal{E}$. $|\mathcal{E}|$ denotes the number of edges. The assortativity $\alpha$ lies in the range of [-1, 1], where a positive $\alpha$ indicates that high degree nodes have high probabilities of linking to other nodes with high degrees on average. In contrast, when $\alpha$ is negative, it is a disassortative network and the high-degree nodes are more likely to link to low-degree nodes. More specifically, when $\alpha = 0$, we say the network is neutral, and neither the tendencies of linking to high-degree nor low-degree nodes are observed. Figure 4a shows that the assortativity of NFT transaction network is negative in the recent six years, and it increases year by year, gradually approaching to zero. This indicates that the emerging transaction network is evolving from disassortative to assortative, which means there are more and more hub nodes available for themselves to connect to each other to increase the assortativity. A detailed analysis is given in Section 4.2.

**Density** calculates the ratio of existing edges over the number of possible edges [49]. The density $d$ is 1 for a complete network, and it can be larger than 1 when self-loops or multi-edges are taken into consideration. We compute the density of a directed network as follows:

$$d = \frac{|\mathcal{E}|}{|\mathcal{V}|(|\mathcal{V}| - 1)} \tag{2}$$

where $|\mathcal{E}|$ indicates the number of edges and $|\mathcal{V}|$ represents node numbers. As we can see in Figure 4b, the density drops rapidly, which indicates the network becomes sparser over time. The decrease of density is mainly caused by the increment of edges are less than the node increment. It also means that the network utilization is quite low and the interactions between different nodes are very limited (i.e., one node only interacts with a few other nodes). This is understandable, since creating a account (i.e., node) is free, but making transactions (i.e., creating edges) cost gas fees. This property is quite different from citation networks [41], where density becomes denser over the time.

**Reciprocity** in a directed network is determined by the proportion of bidirectional edges to the number of total edges [67, 3]. Formally, the reciprocity $r$ is calculated as:

$$r = \frac{\sum \mathbb{I}((i,j) \in \mathcal{E} \bigwedge (j,i) \in \mathcal{E}))}{|\mathcal{E}|} \tag{3}$$

where $|\mathcal{E}|$ indicates the number of edges. $\mathbb{I}(\cdot)$ is an indicator function and it returns 1 when node $i$ and $j$ have bidirectional edge, otherwise it returns 0. The trend of reciprocity is demonstrated in Figure 4c. The relation between reciprocity and time is not monotonous. In general, it increases at the first, and then decreases in the following several years. This may be due to the emerging of NFT swapping, which allows users to swap their NFTs with each other. However, as time goes on, the NFT market becomes more and more mature, and users are prone to trading instead of swapping their NFTs to gain more profits.

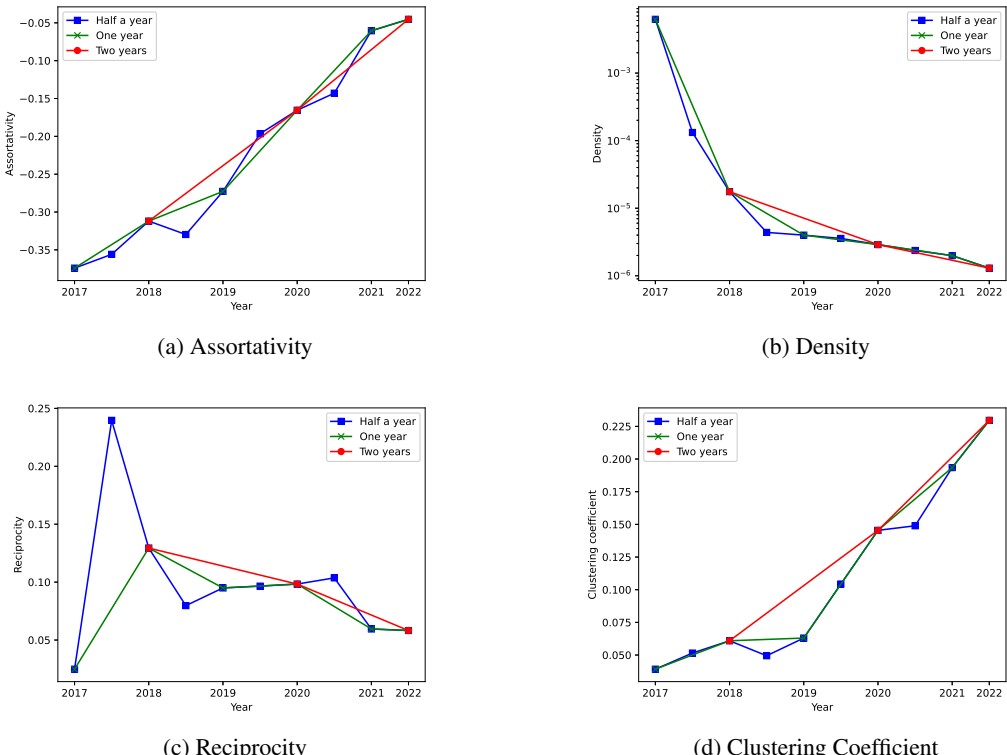

(a) Assortativity

(b) Density

(c) Reciprocity

(d) Clustering Coefficient

Figure 4: Evolution of network global properties with different time granularity.

**Average Clustering Coefficient** evaluates to what extend the nodes in a network tend to tightly cluster together [61], which is computed by averaging local clustering coefficient across all nodes. Node $v_i$'s local clustering coefficient $c_i$ is the percentage of edges among its neighborhood divided by all the possible edges between its neighborhood. We formulate the local clustering coefficient for directed network as:

$$c_i = \frac{|\{e_{jk} : v_j, v_k \in \mathcal{N}_i, e_{jk} \in \mathcal{E}\}|}{|\mathcal{N}_i|(|\mathcal{N}_i| - 1)} \tag{4}$$

where $\mathcal{N}_i = \{v_j : e_{ij} \in \mathcal{E} \bigvee e_{ji} \in \mathcal{E}\}$ is the neighborhood of node $i$. Edge $e_{jk}$ indicates the link between node $j$ and node $k$. Thus, the equation for the average clustering coefficient is as follows:

$$c = \frac{1}{|\mathcal{V}|} \sum_{i=1}^{|\mathcal{V}|} c_i \tag{5}$$

where $|\mathcal{V}|$ denotes the number of nodes. In Figure 4d, we present the clustering coefficient for the NFT transaction network across six years. We can observe that the average clustering coefficient is growing stably, which indicates the network is forming an increasing number of tightly connected clusters or communities. One possible explanation is that those accounts in a tightly connected clusters are actually controlled by the same person, and they utilize multiple accounts to conduct money laundering and wash trading [73], etc.

Furthermore, Figure 4 also illustrates the trends of different properties under different time granularities. In particular, the networks constructed with different time granularities could highlight the anomalies during its evolving procedure. Here, the anomaly means a property value that is much more larger or smaller than the value in its neighborhood time periods. For instance, the reciprocity in Figure 4c indicates that its value in the middle of 2018 is twice larger than that of the other time period. Since this abnormal value is observed in a half-year granularity, we can dive into a finer time scale, which is 3-month and monthly time scopes. The results are illustrated in Figure 5. As we can see in Figure 5a, the reciprocity values are significantly different between the first and the second

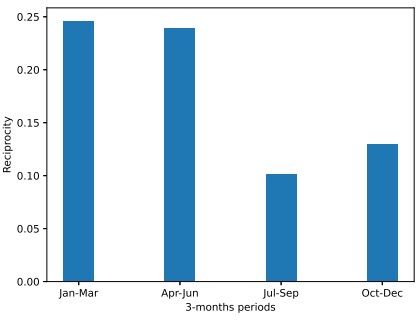
(a) 3-months granularity of year 2018

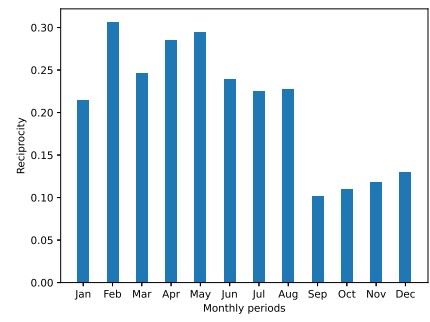
(b) Monthly granularity of year 2018

Figure 5: Finer time granularity analysis on reciprocity.

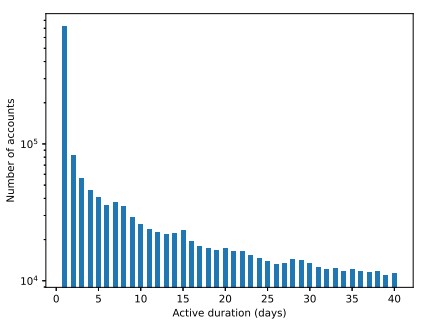
(a) Number of active accounts

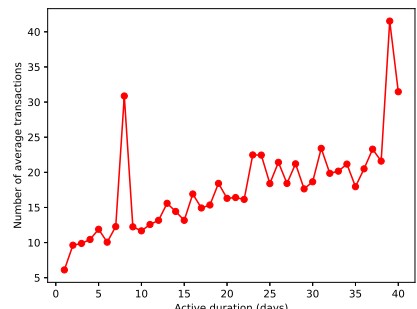
(b) Number of average transactions

Figure 6: Accounts' active periods.

half of year 2018. Meanwhile, we further investigate the data with a monthly basis, which makes it possible for us to locate the specific months. Figure 5b demonstrates that the anomaly is lasting from January to August, and reaches its peak at February. These finer granularity analyses demonstrate that monthly data may be more helpful in the scenario of anomaly detection.

# D    Dynamic Behavior Analyses

## D.1    Active Period of Nodes

Based on our aforementioned analyses, the NFT transaction network is highly active with new nodes continuously adding in. We now proceed to investigate the nodes' active periods. Here, the node's active period is defined as the time interval between its first transaction and its last transaction. In this case, we discard those nodes with only one transaction in the dataset. Figure 6 presents the statistical information for node's active duration in the scale of days. Note that, we only show the active periods from 1 day to 40 days in Figure 6a. According to our data, 23.43% of the nodes have the active period of 1 day, and up to 47.25% nodes' active periods are 1 month. This is consistent with our previous observations regarding the highly increasing number of edges. It's worth noting that only one account has the active period of nearly 5 years. After checking the data, we find out that this node is associated with the Null address (i.e., *0x000...000*). It is not a surprise, since all NFT mint activities would build connections with the Null address. This also indicates that there exist new NFT tokens being continuously minted, which reveals the fast growing speed of NFT market.

Then, we explore the impact of a node's active period on the number of transactions it engages in. One natural hypothesis is that the longer the node's active period, the more transactions it will create, which is consistent with the user behaviours in social networks. Figure 6b shows the number of average transactions with different active periods. As can be seen, the active period and its average

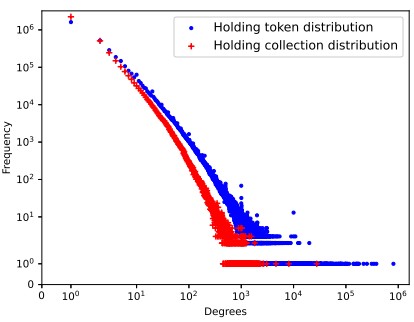

(a) Distributions of tokens and collections

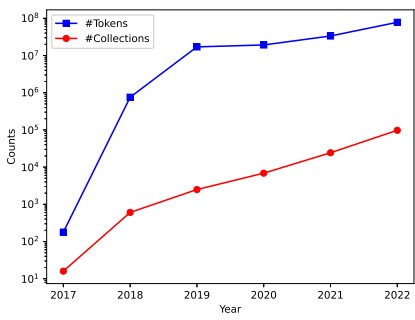

(b) Number of tokens and collections

Figure 7: Distributions of tokens and collections.

transactions are positively correlated. The number of transactions increase exponentially as the active periods become longer. We also notice two spikes with active periods of 8 days and 39 days. Among them, address *0x610...662*[7] causes the spikes of average transactions in active periods of 8 days, which is a Ethereum Name Service (ENS) migration contract to migrate second-level names from the old registry and registrar to the new ones. It creates more than 685,000 transactions within 8 days. To summarize, even through there are some addresses with rather limited active periods, we can generally conclude that most of the addresses are quite active with lots of transactions.

### D.2 Holding Tokens and Collections

To reveal the characteristics of the NFT economy, we analyse the distributions of holding tokens and collections for different accounts. Formally, we trace all the received tokens and sent tokens for each account. Through this way, we can know the owner of each token and the quantity of tokens held by an account. Figure 7a shows that both the accounts' holding tokens and collections are power law distributions. Specifically, 42.10% of the accounts hold only one token, and 58.50% of the accounts hold tokens from one NFT collection. Moreover, there are about 81.70% accounts holding no more than 10 tokens, and 91.40% of accounts hold tokens from no more than 10 collections. In the middle of year 2022, the largest "holder" is the Null address (i.e., *0x000...000*), which involves 809,125 tokens from 8,126 collections. Note that the Null address is a special account, and sending tokens to Null address means destroying the tokens whereas receiving tokens from Null address indicates minting tokens. There are several reasons that people destroy their tokens, including reducing the supply to increase a collection's value or rectifying error information in the tokens. Then, we look at the next valid holding address. The second largest holder is associated with address *0x000...7a2* [8], which holds 381,570 tokens from 1,454 collections. Since it holds so many tokens, we are interested in uncovering its identity. Therefore, we try to uncover all the relevant activities associated with the address *0x000...7a2*. First of all, we search the address in the Ethereum Name Service and find that the address is bound with the name *stronghands.eth*[9]. Then, we search the keyword "stronghands" with Google, and the results show that it is a blockchain community[10], which supports issuing tokens, trading NFTs and bridge integrations, etc.

Table 6 and Table 7 list the top-10 largest holders in the year of 2020 and 2021, respectively. As can be seen, most top-10 holders in 2020 continue to be top-10 holders in 2021 as well. It is interesting to note that the relative positions are almost same, except that address *0xd9a...6a5* and *0x721...ace* swap their positions in 2021, and the Null address becomes the second largest "holder". Among them, address *0xff1...2c8* drops out of the top-10 list in 2021, and address *0xe05...9d5* enters the top-10 list in 2021, which are annotated as bold. The little difference in top-10 list for year 2020 and 2021 indicates that it is very difficult to be one of the top holders for new users, since it costs a lot of money to mint or buy NFTs. Moreover, when looking at the numbers in Table 6 and Table 7, we can find that

---

[7]0x6109DD117AA5486605FC85e040ab00163a75c662

[8]0x0008d343091ef8bd3efa730f6aae5a26a285c7a2

[9]https://app.ens.domains/address/0x0008d343091ef8bd3efa730f6aae5a26a285c7a2

[10]https://www.stronghands.io/

Table 6: Top-10 accounts' holding tokens and collections in 2020.

| Account Address | Tokens | Collections |
|---|---|---|
| 0x0008d343091ef8bd3efa730f6aae5a26a285c7a2 | 363,962 | 36 |
| 0x26cdee4269273e1ea5dfac6b5791df2656897738 | 343,413 | 14 |
| 0xe4a8dfca175cdca4ae370f5b7aaff24bd1c9c8ef | 308,916 | 13 |
| 0xf7ee6c2f811b52c72efd167a1bb3f4adaa1e0f89 | 216,477 | 39 |
| 0x09c1e4c1adad99436b5c22a395174a1320ee716b | 166,321 | 1 |
| 0xf33bd4edc6dcd7240966f20401014ad0018d065b | 161,368 | 19 |
| 0xd9ab699e5e196139b8a1c8f70ead01b2137fc6a5 | 152,788 | 16 |
| 0x721931508df2764fd4f70c53da646cb8aed16ace | 149,115 | 49 |
| 0x0000000000000000000000000000000000000000 | 143,849 | 683 |
| **0xff18298382948028f9d93c4e32be1382204022c8** | 140,025 | 22 |

Table 7: Top-10 accounts' holding tokens and collections in 2021.

| Account Address | Tokens | Collections |
|---|---|---|
| 0x0008d343091ef8bd3efa730f6aae5a26a285c7a2 | 378,893 | 198 |
| 0x0000000000000000000000000000000000000000 | 365,565 | 1,914 |
| 0x26cdee4269273e1ea5dfac6b5791df2656897738 | 343,413 | 14 |
| 0xe4a8dfca175cdca4ae370f5b7aaff24bd1c9c8ef | 308,880 | 13 |
| 0xf7ee6c2f811b52c72efd167a1bb3f4adaa1e0f89 | 217,172 | 83 |
| 0x09c1e4c1adad99436b5c22a395174a1320ee716b | 166,321 | 1 |
| **0xe052113bd7d7700d623414a0a4585bcae754e9d5** | 163,499 | 1,467 |
| 0xf33bd4edc6dcd7240966f20401014ad0018d065b | 161,413 | 20 |
| 0x721931508df2764fd4f70c53da646cb8aed16ace | 159,464 | 204 |
| 0xd9ab699e5e196139b8a1c8f70ead01b2137fc6a5 | 152,788 | 16 |

the top holders' tokens and collections are rapidly growing. One exception is the address *0x09c...16b*, which holds 166K tokens from the same collection (i.e., DozerDoll, a game dirven NFT). On average, they hold 10K more tokens in 2021 compared with the year 2020. This is because there are more and more tokens as well as collections. Figure 7b also verifies this observation. Take the year of 2021 as an example, it increases 17,428 collections and 14 million tokens. Thus, the whole NFT ecosystem is still in its bull market.

## D.3 Evolution of Diameters

We study the diameter of the NFT transaction network in this section, which reflects the communication efficiency among different nodes. Generally, the network's diameter is defined as the largest shortest path among all pairs of nodes in the network. As pointed out by [41, 42], this metric is very sensitive to the noise in the network. For instance, a single long path would result in a large diameter. Thus, we resort to the *effective diameter* used in [41], which is defined as the 90-th percentile of the shortest path length among all pairs of nodes. Figure 8 presents the diameters as it evolves over time by years. The blue line shows the diameter calculated with the whole transaction network, which includes the special Null address. In contrast, the red line indicates the diameter computed without the Null address, which means all the NFT minting and destroying activities are removed from the network. We can observe that these two lines show totally different trends, i.e., one is increasing and the other is decreasing. The final diameter is about 3.0 in 2022 when including the Null address, and it is only half of the value when removing the Null address. It is surprised to see that the Null address has such large influence on this property. With in-depth analysis, we find that there are about 3.5 million addresses connecting with the Null address, which accounts for 77.3% of the total addresses. Thus, the Null address is a huge hub node, and provides a shortcut for nodes to reach each other. As a result, the diameter is smaller in this situation, and every non-mint action would increase its value when the network expands.

To eliminate the effect of the Null address, we focus on the analysis of diameter without Null address, which reveals the transaction behavior patterns without mint. Similar to citation networks that the diameter is shrinking observed by Leskovec et al [41], our results show that the diameter also shrinks in NFT transaction network, i.e., decreasing from 5.94 to 4.72. Then, we explore the graph structure to see whether we can explain the diameter shrinking phenomenon. Specifically, we observe that

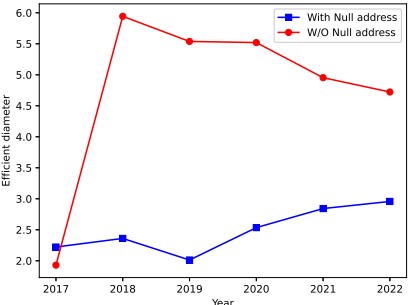

Figure 8: Effective diameter by years.

about 42.28% of the nodes have the degree of 1 in the final network. Those nodes with degree 1 are the key factor that leads to the increase of diameter. Thus, we remove these degree 1 nodes, and calculate the effective diameter of the remaining part. The value is 4.20, which shrinks again. After that, we repeat this process again, and find that it only has 2.26% nodes with degree 1 in the remaining part. Similarly, we delete those degree 1 nodes, and compute its effective diameter again, which is 4.28 and stays the same as before. This suggests that there is a giant component with extremely high connectivity, and nodes with degree 1 is only a thin layer at the outside of the well-connected giant component. This observation brings new challenges for some downstream tasks.

## E More Analyses on Continuous Subgraph Matching

**Frameworks.** We evaluate the performance of six recent CSM frameworks in this section. (1) SJ-Tree [13] proposes a lazy search algorithm, and the search strategy is determined by a vertex-to-vertex basis, which depends on the likelihood of a matching in the vertex neighborhood. (2) IEDyn [30, 31] randomly selects a node from query graph, then it products matching order by conducting DFS on the query graph. (3) SymBi [50] employs a dynamic candidate space as an auxiliary data structure for filtering. (4) Graphflow [33] first generates matching order offline, and then retrieves it in online processing. (5) TurboFlux [35] employs a concise representation of the intermediate results, and a novel edge transition model is proposed to identify the update operations that may affect the current solutions. (6) RapidFlow [68] performs batch subgraph matching via designing a query reduction technique, then dual matching is utilized to leverage the duality of the graph in the matching procedure.

**Settings.** Similar to the link prediction task, we also remove all the transactions related to the Null address, which gets rid of the impact of the extreme large degree node. In previous studies [69, 35, 13], the initial graph is constructed using the first 90% of edges, while the rest 10% of edges are employed as insertion streams. In our scenario, we know the exact time of each transaction, thus we use NFT transactions from year 2017 to the end of 2021 as the initial graph, and then the transactions in the year of 2022 are regarded as the insertion streams. Since the original nodes and edges do not have fine-grained labels, each node is assigned a label randomly selected from a pool of 30 labels. We do not assign labels for edges following the previous works [69, 35]. For evaluation metrics, we report the query time, which denotes the time taken by the online matching procedure to execute. We discard the graph update time, since it's same for all the frameworks. To complete the experiments under an affordable time, a one-hour time limit is imposed for query processing (i.e., $3.6 \times 10^6$ ms). Additionally, we also calculate the number of matched subgraphs for each query graph.

**Results.** As we have discussed in Section 4.2, there exist lots of hub nodes in the NFT transaction network. Therefore, we are interested in whether updating the index of hub nodes will have a great impact on the query time. Specifically, except Graphflow, all the remaining frameworks belong to the index-based incremental computation. In this context, the query time consists of the index time, which signifies the time spent on updating the index, and the enumeration time that enumerates the matched results. Therefore, we first sort the node degrees in the descending order, and the results show that the top-50 largest hub nodes have the largest degree with 433,114 and the smallest degree

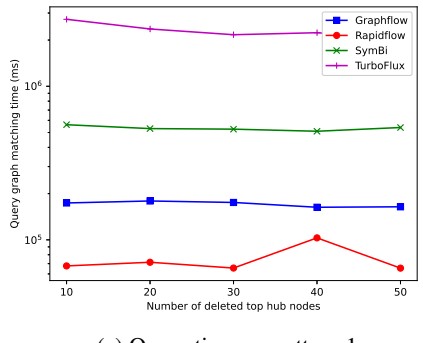

(a) Query time on pattern 1

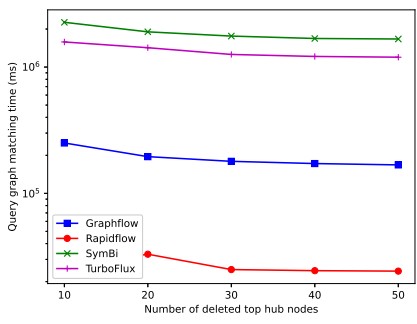

(b) Query time on pattern 3

Figure 9: Query time with deleted hub nodes.

Table 8: Temporal link prediction results with live-update settings in sampled subset. We repeat experiments with 3 random seeds to report the mean as well as standard deviation of AUC and MRR. We also present the results under different time snapshot granularities, e.g., days, weeks and months.

| Models | Snapshot Days | | Snapshot Weeks | | Snapshot Months | |
|---|---|---|---|---|---|---|
| | AUC | MRR | AUC | MRR | AUC | MRR |
| Dyngraph2vec | 77.39±2.04 | 40.29±5.75 | 75.33±7.15 | 36.84±4.97 | 72.03±2.73 | 34.83±4.20 |
| TGCN | 95.49±0.78 | 62.40±7.89 | 91.09±0.46 | 41.69±7.82 | 86.13±1.33 | **43.67±6.23** |
| EvolveGCN | 84.34±2.81 | 40.91±7.96 | 79.13±0.92 | 40.72±3.82 | 78.40±3.33 | 42.86±3.71 |
| GCRN-GRU | 91.41±1.24 | 33.65±9.48 | 90.78±2.43 | 42.88±0.48 | 82.64±0.22 | 38.99±0.18 |
| GCRN-LSTM | 93.97±1.66 | 41.63±3.22 | 90.16±1.83 | 39.20±7.19 | 83.38±0.71 | 37.22±2.63 |
| DynGEM | 95.67±0.74 | 49.25±6.01 | 93.34±0.98 | 45.41±5.29 | 90.75±1.29 | 41.38±0.61 |
| Roland-MA | **97.55±0.59** | 37.18±6.76 | **94.59±0.50** | 44.26±1.26 | 87.69±2.17 | 37.98±4.61 |
| Roland-MLP | 96.38±1.07 | 60.44±10.0 | 89.25±0.89 | 52.32±3.20 | 87.89±0.96 | 42.02±3.37 |
| Roland-GRU | 96.52±0.08 | **69.34±2.68** | 92.83±0.82 | **52.74±2.60** | **91.01±0.77** | 41.59±2.02 |

with 7,623. Then, we delete the top-50 hub nodes sequentially and report the query time in Figure 9 for pattern 1 and pattern 3. Note that, we remove the randomly assigned node labels in this situation, which counts the query time associated with the hub nodes. As can be seen, the query time almost remains the same with acceptable fluctuations across all the frameworks except for RapidFlow. This is because RapidFlow is extremely efficient (i.e, only taking several hundred milliseconds), thus a slight fluctuation will be very obvious in the figure. We can conclude that these four frameworks are applicable to graphs with dense structures. They can effectively serve as a bridge to generate ground truth for continuous subgraph matching, providing valuable support for the training of deep graph learning models.

## F  Temporal Link Prediction Results on Sampled Subset

To fully evaluate the models' performance, we also sample a subset of data spanning from January 1st 2020 to August 31st 2020. This subset contains 88,112 nodes and 203,221 edges. We provide the temporal link prediction results under live update setting in Table 8. As can be seen from the results, we can draw similar conclusions. Notably, we also observe that with the increase in time granularity, the performance of the models exhibits a corresponding decline within this sub-sampled dataset.

## G  TEA and TET Plots

Following the definitions in [57], we present Temporal Edge Appearance (TEA) and Temporal Edge Traffic (TET) plots in Figure 10. Specifically, a TEA plot visualizes the proportion of recurring edges compared to newly joined edges at each timestamp within a temporal graph. In Figure 10a, the gray bar represents the count of edges that are previously observed, while the red bar signifies the quantity of new edges generated at each subsequent time step. The TEA plot illustrates that our

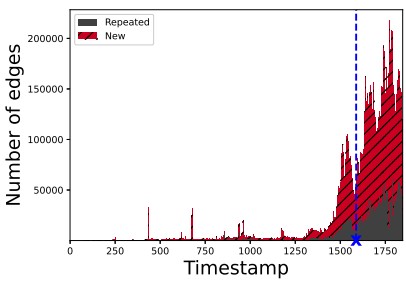

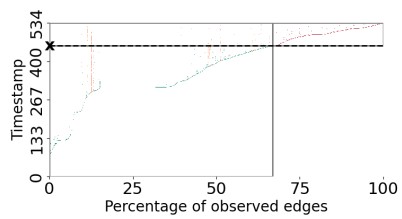

(a) TEA plot with day intervals      (b) TET plot with sampled 2 million edges

Figure 10: TEA and TET plots.

dataset exhibits a substantial proportion of newly formed edges at each time step, which means the transaction graph is highly active. In contrast, a TET plot represents the recurrent pattern of edges across different time intervals. Edges are colored to indicate whether they appear solely in the training set (green), solely in the test set (red), or in both sets (orange). As it is time consuming to plot all the 124 million edges in the figure, we randomly sample 2 million edges and show the results in Figure 10b. We have tried to use all the data, but the plotting process was unable to complete within 12 hours. The sampled 2 million edges subset showcases a subtle recurrence pattern, which is consistent with our analyses in previous sections.

