# OpenReview forum: "Live Graph Lab: Towards Open, Dynamic and Real Transaction Graphs with NFT"
_NeurIPS.cc/2023/Track/Datasets_and_Benchmarks — NeurIPS 2023 Datasets and Benchmarks Poster_

### Official Review · Reviewer_RoMo · 2023-07-02
**A useful data set but lacks of novelty**

**Rating:** 5
**Confidence:** 4
**Clarity:** This paper is well-written and easy t…

**Strengths:**

- Data construction. The construction of the dataset is well documented.
- The dataset is well-analyzed and useful to understand the NFT ecosystem.

**Additional Feedback:**

N/A

**Correctness:**

- When mentioning related graph datasets in section 2, the paper claims "Although these datasets are publicly available, they are not constantly updated as time goes by". This statement might be too strong. For example, AMiner generates the datasets from the citation network, apparently, the citation network is constantly updated as time goes by, and AMiner indeed keeps its network updated.

**Documentation:**

- The procedure to generate the dataset is well documented in the paper.
- The code for generating the dataset is not found in the provided code repo.
- The code of the graph analysis and link prediction is provided and well documented.

**Ethics:**

No ethical concerns.

**Limitations:**

- More discussion on the unique role in advancing the machine learning method, benefiting the NeurIPS community, and the societal impact is highly recommended.

**Opportunities For Improvement:**

- Originality: To understand the ecosystem, the transaction graph of blockchains has been well studied for various applications in previous literature, including the evolution of the transaction in Ethereum from a temporal graph perspective, and temporal analysis of the entire Ethereum blockchain network. However, the previous works have used the same data source with similar data processing steps.

- Impact and relevance to the NeurIPS: The dataset introduced in the paper is useful to understand the blockchain ecosystem, however, the NeurIPS community has a wider interest in information processing systems other than a specific problem. Although this paper shows the use of the dataset in Link prediction, it lacks a comparison with other similar datasets and its unique role in driving machine learning tasks.

- Reproducibility: I thank the paper for providing the code and the data for the experiments. However, the code for generating the graph from Ethereum is not found in the provided repo.

**Relation To Prior Work:**

- Lack of import references in the section of related datasets. There are several graph datasets are mentioned in the related datasets section, but there are other important works using the same blockchains/Ethereum transaction as temporal graphs are missing.

**Summary And Contributions:**

This paper introduces a temporal graph extracted from the transaction of blockchains - namely the Non-fungible token (NFT) transaction in the Ethereum blockchain.

The paper describes the data downloading and parsing process, analyzes the structure and dynamics of the temporal graph using a series of measurements, and reveals the properties of the NFT ecosystem. The paper also applies some Graph Neural Network (GNN) related models on this dataset for the downstream application of Temporal Link Prediction.

The data and code involved in the analysis and experiments are also provided.

---

> ### Author Response · Authors · 2023-08-22
> **Response to Reviewer RoMo (Part I)**
>
> We sincerely thank the reviewer RoMo’s feedback. Our detailed responses to your concerns are listed as follows:
>
> *Q1. However, the previous works have used the same data source with similar data processing steps.*
>
> A1. Thanks for pointing this out. We acknowledge that previous research has indeed explored data source from the Ethereum ecosystem. It’s important to note that these previous works mainly focus on fungible token or ETH transactions. However, Non-Fungible Tokens (NFTs), which are a vital component of the Ethereum blockchain, have been overlooked by existing works. Given its non-fungible property, we can trace the flow of tokens and explore other potential tasks, such as predicting the next token a user might purchase, etc. Moreover, our proposed Live Graph Lab provides datasets that have two unique characters absent in existing datasets: 1) it has complete structure, and 2) it is constantly evolving in a real-time manner.
>
> ***
> *Q2. Although this paper shows the use of the dataset in Link prediction, it lacks a comparison with other similar datasets and its unique role in driving machine learning tasks.*
>
> A2. Thanks for pointing this out. As we have described in the paper, our dataset has two unique characters: it is constantly evolving in a real-time manner, and it is complete. As demonstrated in Table 1, currently widely used datasets do not have these properties. However, these characters are important for various downstream applications. For instance, if the graph is incomplete or their characteristics have changed significantly, the learning outcomes could be ineffective and even misleading in the graph learning tasks. Moreover, our dataset is large-scale with 4.5 million nodes and 124 million edges spanning several years. We also demonstrate the necessity of modeling the highly active NFT transaction network with a finer time granularity.
>
> ***
> *Q3. I thank the paper for providing the code and the data for the experiments. However, the code for generating the graph from Ethereum is not found in the provided repo.*
>
> A3. Thanks for pointing this out. As mentioned in Section 3.2, we use Ethereum-ETL (https://github.com/blockchain-etl/ethereum-etl) to parse the Ethereum transactions. Then, you can use any tool you like to construct the graph such as NetworkX or igraph. We have done these tasks for your convenience, so you won't have to engage in the laborious parsing work. We have updated our website (https://livegraphlab.github.io) with the live component. At this stage, we are offering a prototype version and our ongoing efforts are focused on refining and enhancing the user experience. Our goal is to present this tool in a manner that is not only user-friendly but also stable and reliable.
>
> ***
> *Q4. More discussion on the unique role in advancing the machine learning method, benefiting the NeurIPS community, and the societal impact is highly recommended.*
>
> A4. Thanks for pointing this out. We provide more discussions here.
>
> **Role in the Community**. The Live Graph Lab can facilitate researchers from graph community by offering comprehensive blockchain based graphs via an easily accessible manner. In the current literature, studies are usually conducted on a set of outdated and incomplete graphs. We propose the concept of Live Graph Lab, which provides live graphs according to blockchain transactions. It not only alleviates the researchers’ burden of parsing massive blockchain data, but also brings a considerable of opportunities to conduct experiments in the real-world scenario for temporal graph studies.
>
> **Societal Impact**.
> The Live Graph Lab can facilitate researchers from graph community by offering comprehensive blockchain based graphs via an easily accessible manner. Insights from this research can be directly applied to improve the design, security, and user experience of NFT platforms, leading to sustainable growth of the ecosystem. Meanwhile, as the live graph constantly records all the NFT transactions in the Ethereum blockchain, the possibility of encountering malicious activities could become a concern, such as bot transactions or wash trading, etc. It might also cause potential negative societal impacts. Since the dataset consists of complete transactions associated with the wallet addresses, it could enable the tracking of each wallet’s behaviors, habits, and financial activities. This kind of tracking could be exploited for targeted advertising, manipulation, or surveillance. The dataset could also make it possible for malicious actors to analyze the transaction patterns and manipulate the NFT market, which could lead to unfair practices, price manipulation, and market instability.

---

> > ### Author Response · Authors · 2023-08-22
> > **Response to Reviewer RoMo (Part II)**
> >
> > *Q5. When mentioning related graph datasets in section 2, the paper claims "Although these datasets are publicly available, they are not constantly updated as time goes by". This statement might be too strong. For example, AMiner generates the datasets from the citation network, apparently, the citation network is constantly updated as time goes by, and AMiner indeed keeps its network updated.*
> >
> > A5. Thanks for pointing this out. We have revised the corresponding descriptions. We actually mean these datasets are not constantly evolving in a timely manner. Sorry for causing the confusion.
> >
> > ***
> > *Q6. Lack of import references in the section of related datasets. There are several graph datasets are mentioned in the related datasets section, but there are other important works using the same blockchains/Ethereum transaction as temporal graphs are missing.*
> >
> > A6. Thanks for pointing this out. We have added more discussion in the Section 2. For more details, please refer to the revised paper.

---

> ### Author Response · Authors · 2023-08-26
> **A gentle reminder for discussion**
>
> Dear Reviewer RoMo,
>
> We would like to express our sincere gratitude for the time and effort you have put into reviewing our paper and providing us with valuable feedback. We have tried our best to address your concerns.
>
> We kindly request that you take a few moments to review our rebuttal and let us know if there are any further concerns that we can address. We would appreciate your prompt response.
>
> Thank you once again for your feedback.

---

> > ### Comment · Reviewer_RoMo · 2023-08-28
> >
> > Thanks for the response, which cleared up my previous concerns. I would like to increase my score.

---

> > > ### Author Response · Authors · 2023-08-29
> > > **Thanks for your reply!**
> > >
> > > We sincerely appreciate your feedback and thanks for increasing the score. If there are any further concerns that you'd like us to address, please let us know.
> > >
> > > Thanks again for your response.

---

### Official Review · Reviewer_dqJb · 2023-07-21
**A new dataset with comprehensive analysis**

**Rating:** 6
**Confidence:** 3
**Correctness:** N/A

**Strengths:**

1. A large-scale real-world dynamic dataset with rich tasks and properties that could benefit blockchain and graph machine learning research.

2. Comprehensive analysis of the graph properties and various graph-based machine learning tasks.

3. A well-documented dataset repository.

**Additional Feedback:**

Suggestions for improvement:
1. In the main body of the paper, the authors should present as many insights and highlights of the findings as possible. Sections 4 and 5 include too many details which could be committed or put into the appendix. The main paper needs to summarize the analytical findings and benchmarking performance of other tasks in the appendix.
2. As a dataset/benchmark paper, the authors should discuss more about the future directions and potential use cases of the proposed dataset.
3. Lines 156-161, those findings and discussions are too trivial and lack reference. If the authors are not blockchain/NFT experts, I suggest that authors focus more on showing the properties of the graph instead of making too many arguments and conclusions.
4. Line 136, missing the reference for the majority of existing studies.
5. The CSM task is pretty interesting and should be highlighted in the paper.


Questions:
1. What is the node feature in the temporal link prediction task?
2. Why are bi-directional edges considered abnormal or suspicious? Is there any evidence/previous work supporting this argument?
3. Why are the hub nodes with the specific patterns bots? Are bots common in blockchain? Any literature discussing bots in Ethereum?
4. What are the sources and reasons for the five most common washing trading patterns shown in Fig.8 in Appendix?
5. In Fig. 8, pattern 3 is a subgraph of pattern 5. Why are there more pattern 5 counts than pattern 3 counts in the graph in Table 7?
6. For the node classification task in the Appendix, I don't see the necessity to train a GNN model to predict the node label since the label can be calculated based on an account's trading pattern. The node label should be a property correlated to graph and node features but cannot be directly derived.

Typos and grammar issues:
1. Line 317, utilize -> utilizes
2. Line 278, delete recent.
3. Line 10, know -> knowledge.

**Clarity:**

The paper structure can be improved and there are some typos. Please see the details in additional feedback.

**Documentation:**

The paper misses the ethical statement and the dataset license. The data collection and graph construction descriptions are generally complete.

**Ethics:**

No ethical concern since the data is publicly available, and no personal information is included.

**Limitations:**

1. The authors claim that the Live Graph Lab can periodically collect the latest NFT transactions, but I could not find any live component in the given website.

2. Some of the arguments and conclusions are unfounded and need to be elaborated with more evidence.

**Opportunities For Improvement:**

Please see additional feedback at the end.

**Relation To Prior Work:**

The literature review is comprehensive and clearly differentiates this work from previous works.

**Summary And Contributions:**

This submission proposes a dynamic graph dataset composed of open-source blockchain accounts as nodes and NFT transactions as edges. The authors analyze the property of the graph and present some insights. Different graph-related tasks, including node classification, link prediction, and subgraph matching, are evaluated based on the new datasets.

---

> ### Author Response · Authors · 2023-08-22
> **Response to Reviewer dqJb (Part I)**
>
> We sincerely thank the reviewer dqJb for the constructive feedback. Our detailed responses to your concerns are listed as follows:
>
> *Q1. Sections 4 and 5 include too many details which could be committed or put into the appendix. The main paper needs to summarize the analytical findings and benchmarking performance of other tasks in the appendix.*
>
> A1. Thanks for pointing this out. We have reorganized our paper’s structure. For more details, please refer to the revised paper. We have utilized blue color to highlight the modifications.
>
> ***
> *Q2. As a dataset/benchmark paper, the authors should discuss more about the future directions and potential use cases of the proposed dataset.*
>
> A2. Thanks for pointing this out. We have added more discussion about the future directions in the conclusion section. For more details, please refer to the revised paper.
>
> ***
> *Q3. Lines 156-161, those findings and discussions are too trivial and lack reference. If the authors are not blockchain/NFT experts, I suggest that authors focus more on showing the properties of the graph instead of making too many arguments and conclusions.*
>
> A3. Thanks for this suggestion. We have made revisions based on your feedback. For more details, please refer to the revised paper.
>
> ***
> *Q4. The CSM task is pretty interesting and should be highlighted in the paper.*
>
> A4. Thanks for pointing this out. We have reorganized our paper’s structure. For more details, please refer to the revised paper. We have utilized blue color to highlight the modifications.
>
> ***
>
> *Q5. The authors claim that the Live Graph Lab can periodically collect the latest NFT transactions, but I could not find any live component in the given website.*
>
> A5. Thanks for pointing this out. We have updated our website (https://livegraphlab.github.io) with the live component. At this stage, we are offering a prototype version and our ongoing efforts are focused on refining and enhancing the user experience. Our goal is to present this tool in a manner that is not only user-friendly but also stable and reliable.
>
> ***
> *Q6. What is the node feature in the temporal link prediction task?*
>
> A6. Thanks for pointing this out. We set node feature as 1 in the temporal link prediction task. However, our datasets offer the flexibility to incorporate diverse node features. For instance, we can construct node features by utilizing information such as the total incoming transaction value, total outgoing transaction value, the count of held tokens, etc.
>
> ***
>
> *Q7. Why are bi-directional edges considered abnormal or suspicious? Is there any evidence/previous work supporting this argument?*
>
> A7. Thanks for pointing this out. Not all the bi-directional edges are considered abnormal or suspicious. In the context, we mean the **simultaneously** formed bi-directional edges can be a good indicator for identifying the abnormal activities. As indicated by references [1,2], the mutual interactions account for a significant portion of wash trading activities.
>
> [1] von Wachter V, Jensen J R, Regner F, et al. NFT Wash Trading: Quantifying Suspicious Behaviour in NFT markets[C]//Financial Cryptography and Data Security. FC 2022 International Workshops.
>
> [2] Victor F, Weintraud A M. Detecting and quantifying wash trading on decentralized cryptocurrency exchanges[C]//Proceedings of the Web Conference 2021. 2021: 23-32.
>
> ***
> *Q8. Why are the hub nodes with the specific patterns bots? Are bots common in blockchain? Any literature discussing bots in Ethereum?*
>
> A8. Thanks for pointing this out. We say the specific hub node has high probability of being a bot, because it creates a transaction every a few minutes and the IDs of transferred tokens are in a continuous order. Bots are indeed present in Ethereum ecosystems, and they can have different purposes such as **MEV bot** which runs algorithms to detect and execute profitable MEV opportunities on the Ethereum blockchain, and **Sniper bot** which automates the process of sniping, or placing a last-second bid on an auction item, etc. References [3,4] are several relevant literatures.
>
> [3] Weintraub B, Torres C F, Nita-Rotaru C, et al. A flash (bot) in the pan: measuring maximal extractable value in private pools[C]//Proceedings of the 22nd ACM Internet Measurement Conference. 2022: 458-471.
>
> [4] Daian P, Goldfeder S, Kell T, et al. Flash boys 2.0: Frontrunning in decentralized exchanges, miner extractable value, and consensus instability[C]//2020 IEEE Symposium on Security and Privacy (SP). IEEE, 2020: 910-927.

---

> > ### Author Response · Authors · 2023-08-22
> > **Response to Reviewer dqJb (Part II)**
> >
> > *Q9. What are the sources and reasons for the five most common washing trading patterns shown in Fig.8 in Appendix?*
> >
> > A9. Thanks for pointing this out. We define the wash trading patterns by utilizing wash trading’ definition as well as the findings provided in reference [1,2]. Wash trading is an activity where the seller is on both sides of the trade. The more complex the structures, the less frequently they appear. Based on the above information, we first identify wash trades according to the information provided in https://dune.com/qboy29/looksrare-wash-trades, which treats "back-and-forth trades", very high sales and abnormal rewards as wash trades. Then, we sort the wash trades in descending order of their frequency and selected the top five patterns.
> >
> > [1] von Wachter V, Jensen J R, Regner F, et al. NFT Wash Trading: Quantifying Suspicious Behaviour in NFT markets[C]//Financial Cryptography and Data Security. FC 2022 International Workshops.
> >
> > [2] Victor F, Weintraud A M. Detecting and quantifying wash trading on decentralized cryptocurrency exchanges[C]//Proceedings of the Web Conference 2021. 2021: 23-32.
> >
> > ***
> >
> > *Q10. In Fig. 8, pattern 3 is a subgraph of pattern 5. Why are there more pattern 5 counts than pattern 3 counts in the graph in Table 7?*
> >
> > A10. Thanks for pointing this out. The counts of patterns depend not only on the structure but also on the node labels. It's important to note that Pattern 3 is not necessarily a subgraph of Pattern 5 when considering node labels.
> >
> > ***
> >
> > *Q11. For the node classification task in the Appendix, I don't see the necessity to train a GNN model to predict the node label since the label can be calculated based on an account's trading pattern. The node label should be a property correlated to graph and node features but cannot be directly derived.*
> >
> > A11. Thanks for pointing this out. Our experiments aim to showcase the capabilities of GNNs in handling node classification task under this fast-evolving transaction graph. This is the potential use cases of the proposed dataset. There are also various other potential use cases, such as identifying whether an account holds a specific type of tokens or predicting the range of tokens that the account possesses, etc.

---

> ### Author Response · Authors · 2023-08-26
> **A gentle reminder for discussion**
>
> Dear Reviewer dqJb,
>
> We would like to express our sincere gratitude for the time and effort you have put into reviewing our paper and providing us with valuable feedback. We have tried our best to address your concerns.
>
> We kindly request that you take a few moments to review our rebuttal and let us know if there are any further concerns that we can address. We would appreciate your prompt response.
>
> Thank you once again for your feedback.

---

> ### Comment · Reviewer_dqJb · 2023-08-26
> **Thanks for the response and update**
>
> The authors' response and corresponding addressed most of my concerns. Now, the updated manuscript with more details and references has better readability than the previous one. I would raise my score to 6.
>
> Additional comment: The live graph query page is still rough. The authors should give the timestamp format on the query page so I know how to make a meaningful query. I expect a stable and updated version of this page soon.

---

> > ### Author Response · Authors · 2023-08-27
> > **Thanks for your reply!**
> >
> > Thanks for increasing the score. We will continue to improve our website and ensure it becomes more user-friendly. Thanks for your support!

---

### Official Review · Reviewer_MnCg · 2023-07-21
**Review for Submission 649**

**Rating:** 8
**Confidence:** 4
**Correctness:** yes, the dataset is constructed in a …
**Clarity:** Yes, the paper is well written.

**Strengths:**

As mentioned above, I think that the idea of a live temporal graph dataset / benchmark is an interesting concept, the strength of the submission can be summarized as follows:

- live graph lab, or a live temporal graph dataset presents interesting opportunities for future research and development

- the contributed dataset so far with 4.5 million nodes and 124 million edges spanning years is one of the largest temporal graph datasets available and has interesting dataset properties.

- the experiments on temporal link prediction is extensive and also examined the effect of different time granularities which is often overlooked in the literature.

- the project is well-documented and paper clearly written.

**Additional Feedback:**

Minor Suggestions:

1. table 1 is presented on second page but discussed on Page 3, the paper formatting can be improved

2. plots in figure 1 are too small and hard to see

**Documentation:**

The paper presented clear details on dataset collection. The dataset can be accessed via google drive but can be hosted better on a more permanent link.

**Ethics:**

No ethical concerns.

**Limitations:**

yes, the authors have discussed potential negative societal impacts in the Border Impact and Limitation section.

**Opportunities For Improvement:**

The following are some suggestions for improvement (the suggestions are ranked based on priority):

- the dataset should be hosted on a permanent method other than google drive (as accounts can be lost or deactivated). I would recommend [Zenodo platform](https://zenodo.org/).

- the paper mentions that the data can be extracted live but wasn't clear on how to do that on the website nor in the paper, please add better documentation and an example showing how to collect live graph data say from May 2023 to June 2023?

- The authors discussed in detail the dataset statistics of the collected dataset. It would be interesting to see how it differs from other datasets or even other blockchain transaction graphs, there are multiple datasets available in [Chartalist](https://openreview.net/forum?id=10iA3OowAV3) for example.

- In Figure 1 c). the authors attempt to show statistics on new arriving edges, for edge statistics visualization, if possible, it would be interesting to include the TEA and TET plots from prior work ["Towards Better Evaluation for Dynamic Link Prediction"](https://openreview.net/forum?id=1GVpwr2Tfdg) where it also shows the percentage of repeating edges and novel edges over time and more.

- In Table 3, it is shown that Dyngraph2vec, EvolveGCN and DynGEM are always OOM which shows the problem with method efficiency however, it is difficult to judge how their performance would be given it is OOM. Maybe comparing them in a separate experiment of a subset of the dataset can provide some insights into their performance (if possible).


[update after author response]

The authors have adequately addressed all my concerns, I have thus increased my rating by one. I hope the authors can continue to improve this work and believe this work is a nice contribution to the temporal graph learning community.

**Relation To Prior Work:**

Yes, it is clearly discussed in the paper.

**Summary And Contributions:**

One of the unique aspect of temporal graph learning is that the methods are often designed to adapt in the test set such as seen in [Temporal Graph Network](https://arxiv.org/abs/2006.10637). However, existing datasets are often non-evolving and limited to a fixed period. In this work, the authors presented Live Graph Lab, which evolves in a real-time manner and records all interactions from a blockchain network. I believe this opens up novel opportunities for temporal graph learning methods to compare across longer time horizons and even in real time thus I recommend **accept**. The contributions from the paper are as follows:

- presented the concept of live graph lab, which provides open, evolving and real transaction graphs.

- conducted analysis on a large NFT graph with 4.5 million nodes and 124 million edges. Examined different graph properties and statistics.

- Conducted link prediction experiments utilizing various temporal GNN models across multiple time granularities and showed interesting observations

---

> ### Author Response · Authors · 2023-08-22
> **Response to Reviewer MnCg**
>
> We thank the reviewer MnCg for providing the constructive feedback. We provide the following detailed responses to your concerns.
>
> *Q1. The dataset should be hosted on a permanent method other than google drive (as accounts can be lost or deactivated). I would recommend Zenodo platform.*
>
> A1. Thanks for this suggestion. We have hosted a copy of data on the Zenodo platform. We have updated our website (https://livegraphlab.github.io) and now offered access to the datasets through both Google Drive and Zenodo links (https://zenodo.org/record/8267012).
>
> ***
> *Q2. The paper mentions that the data can be extracted live but wasn't clear on how to do that on the website nor in the paper, please add better documentation and an example showing how to collect live graph data say from May 2023 to June 2023?*
>
> A2. Thanks for pointing this out. We have updated our website (https://livegraphlab.github.io) with the live component. At this stage, we are offering a prototype version and our ongoing efforts are focused on refining and enhancing the user experience. Our goal is to present this tool in a manner that is not only user-friendly but also stable and reliable.
>
> ***
>
> *Q3. It would be interesting to see how it differs from other datasets or even other blockchain transaction graphs, there are multiple datasets available in Chartalist for example.*
>
> A3. Thanks for pointing this out. We want to highlight that our dataset has two unique characters: it is constantly evolving in a real-time manner, and it is complete. As demonstrated in Table 1, currently widely used datasets do not have these properties. However, these characters are important for various downstream applications. For instance, if the graph is incomplete or their characteristics have changed significantly, the learning outcomes could be ineffective and even misleading in the graph learning tasks. Compared with other blockchain transaction graphs, our datasets focus on NFT transactions, an important component in Ethereum blockchain that has been overlooked by current datasets. Given its non-fungible property, we can trace the flow of tokens and explore other potential tasks, such as predicting the next token a user might purchase, etc.
>
> ***
>
> *Q4. If possible, it would be interesting to include the TEA and TET plots from prior work "Towards Better Evaluation for Dynamic Link Prediction" where it also shows the percentage of repeating edges and novel edges over time and more.*
>
> A4. Thanks for pointing this out. Following your suggestion, we include the TEA and TET plots in the Appendix at Section G with Figure 10. Please refer to Appendix for details.
>
> ***
> *Q5. It is difficult to judge how their performance would be given it is OOM. Maybe comparing them in a separate experiment of a subset of the dataset can provide some insights into their performance (if possible).*
>
> A5. Following your valuable suggestion, we have sampled a subset of data spanning from January 1st 2020 to Aug 31st 2020. This subset contains 88,112 nodes and 203,221 edges. We provide the temporal link prediction results under live update setting as follows:
>
> | Methods | Snapshot | Days | Snapshot | Weeks | Snapshot | Months |
> |:--:|:--:|:--:|:--:|:--:|:--:|:--:|
> | Metrics | AUC | MRR | AUC | MRR | AUC | MRR |
> | Dyngraph2vec | 77.39$\pm$2.04 | 40.29$\pm$5.75 | 75.33$\pm$7.15 | 36.84$\pm$4.97 | 72.03$\pm$2.73 | 34.83$\pm$4.20 |
> | TGCN | 95.49$\pm$0.78 | 62.40$\pm$7.89 | 91.09$\pm$0.46 | 41.69$\pm$7.82 | 86.13$\pm$1.33  | 43.67$\pm$6.23 |
> | EvolveGCN | 84.34$\pm$2.81 | 40.91$\pm$7.96 | 79.13$\pm$0.92 | 40.72$\pm$3.82 | 78.40$\pm$3.33 | 42.86$\pm$3.71 |
> | GCRN-GRU | 91.41$\pm$1.24 | 33.65$\pm$9.48 | 90.78$\pm$2.43 | 42.88$\pm$0.48 | 82.64$\pm$0.22 | 38.99$\pm$0.18 |
> | GCRN-LSTM | 93.97$\pm$1.66 | 41.63$\pm$3.22 | 90.16$\pm$1.83 | 39.20$\pm$7.19 | 83.38$\pm$0.71 | 37.22$\pm$2.63 |
> | DynGEM | 95.67$\pm$0.74 | 49.25$\pm$6.01 | 93.34$\pm$0.98 | 45.41$\pm$5.29 | 90.75$\pm$1.29 | 41.38$\pm$0.61 |
> | Roland-MA | 97.55$\pm$0.59 | 37.18$\pm$6.76 | 94.59$\pm$0.50 | 44.26$\pm$1.26 | 87.69$\pm$2.17 | 37.98$\pm$4.61 |
> | Roland-MLP | 96.38$\pm$1.07 | 60.44$\pm$10.0 | 89.25$\pm$0.89 | 52.32$\pm$3.20 | 87.89$\pm$0.96 | 42.02$\pm$3.37 |
> | Roland-GRU | 96.52$\pm$0.08 | 69.34$\pm$2.68 | 92.83$\pm$0.82 | 52.74$\pm$2.60 | 91.01$\pm$0.77 | 41.59$\pm$2.02 |
>
> Based on the results, we can draw similar conclusions outlined in our paper. Notably, we observe that with the increase in time granularity, the performance of the models exhibits a corresponding decline within this sub-sampled dataset.
>
> ***
>
> *Q6. Table 1 is presented on second page but discussed on Page 3, the paper formatting can be improved. plots in figure 1 are too small and hard to see.*
>
> A6. Thanks for pointing this out. We have reorganized the Table 1 and Table 2 to appear on page 3. Additionally, we have increased the text size in Figure 1 and Figure 2 to ensure better clarity. For more details, please refer to the revised paper.

---

> > ### Comment · Reviewer_MnCg · 2023-08-24
> > **Response to Authors**
> >
> > I thank the authors for addressing my concerns in detail and continue to improve their work.
> >
> > - I have tested the live query component on the website, seems to be functional. However inputting timestamps might not be very intuitive to use, I hope in the future version, the authors can provide UI based on selecting which day and or hour of the day.
> > - the observation that time granularity negatively affecting performance is quite interesting, this is a nice addition to the paper.
> > - hosting on zenodo is a welcome addition too.
> >
> > Overall, I believe this work is a nice contribution to the temporal graph learning community and hope the authors will continue to improve it. I will increase my rating by one to reflect that.

---

> > > ### Author Response · Authors · 2023-08-25
> > > **Thanks for your reply!**
> > >
> > > Thank you for increasing the score. We are happy that we have addressed your concerns and we will continue to improve the user interface of our website. Thanks for your support!

---

### Official Review · Reviewer_qE1c · 2023-07-21
**A timely dataset for Blockchain data analytics**

**Rating:** 8
**Confidence:** 5
**Clarity:** The paper is well written.

**Strengths:**

The dataset is not only valuable but also remarkably timely, making a significant contribution to the field of Big Data Analytics (BDA). It presents an excellent opportunity to conduct in-depth studies due to its up-to-date nature. Particularly, the dataset offers valuable insights for conducting essential tasks like link prediction and subgraph matching, which are highly beneficial for advancing research in BDA.

It is also quite remarkable that you parse the data yourself to present a  full coverage.

**Additional Feedback:**

There are a few things on the website that are not mentioned in the paper, such as the CSM data. What is it?

**Correctness:**

- I do not see any issues with the correctness.
- The dataset covers all NFT transactions, evaluations and designs are given in detail.

**Documentation:**

I have checked the web site and analyzed the data files. Everything seems to be in order.

**Ethics:**

I do not see any ethical concerns

**Limitations:**

I do not see any negative societal impact.

**Opportunities For Improvement:**

- The main PDF lacks information on certain tasks, which is not ideal. The website does not give a general figure either. It would be better to move the settings section to the appendix and mention both downstream tasks in this section.
- In Figure 1c, the legend overlaps with the data, which needs to be corrected.
- The role of null address in the minting process should be better explained. Consider adding a figure to illustrate airdrop, minting, etc.
- I found it difficult to understand the significance of the red lines in Figures 2a and 2b.
- I appreciate the authors for citing our Chartalist work, but it's important to note that the networks will be regularly updated, so the sentence in the introduction about them being static is inaccurate.
 -The claim in Fig 8, that these are wash trading patterns.. Which study has found these?

**Relation To Prior Work:**

Yes, the related work covers previous work and previous year's repositories.

**Summary And Contributions:**

The authors instantiate a live graph with an NFT transaction network by synchronizing a full Ethereum node, they present a temporal graph extracted from a specific time period spanning from 2017 to 2022, consisting of over 4.5 million nodes and 124 million edges. Comprehensive analyses of the live NFT transaction network reveal various interesting properties, making it a valuable addition to current datasets and offering exciting opportunities for the graph community.

---

> ### Author Response · Authors · 2023-08-22
> **Response to Reviewer qE1c**
>
> We thank the reviewer qE1c for the detailed and constructive suggestions. We have performed all the changes requested in the minor comments. Our detailed responses to your concerns are listed as follows:
>
> *Q1. The website does not give a general figure either. It would be better to move the settings section to the appendix and mention both downstream tasks in this section.*
>
> A1. Thanks for pointing this out. According to your constructive suggestion, we have updated our website (https://livegraphlab.github.io) with more illustrations. We have also revisited the content in Section 5 and integrated all the downstream tasks into the main paper. For more details, please refer to the revised paper. We have utilized blue color to highlight the modifications.
>
> ***
>
> *Q2. In Figure 1c, the legend overlaps with the data, which needs to be corrected.*
>
> A2. Thanks for pointing this out. We have updated Figure 1 as well as Figure 2 and enlarged their text size for better clarity. For more details, please refer to the revised paper.
>
> ***
>
> *Q3. The role of null address in the minting process should be better explained. Consider adding a figure to illustrate airdrop, minting, etc.*
>
> A3. Thanks for pointing this out. This is a good suggestion. However, distinguishing between an airdrop and mints from null address can be challenging. Airdrops can occur through various mechanisms, including the minting and transferring of tokens, with some involving user claims. Transactions from null addresses further complicate the situation, as it is difficult to definitively determine whether they correspond to an airdrop, a standard mint transaction, or an alternative activity.
>
> ***
>
> *Q4. I found it difficult to understand the significance of the red lines in Figures 2a and 2b.*
>
> A4. Thanks for pointing this out. In Figure 2a and 2b, the blue line indicates the node counts associated the degrees, while the red line represents how many new nodes will connect to nodes of different degrees. As we can see from the figure, if a node has a higher degree (i.e., greater than 100), it will have more new node connections. This is also the reason why the red line initially decreases and then starts to increase. It follows the preferential attachment growth model.
>
> ***
>
> *Q5. It's important to note that the Chartalist networks will be regularly updated, so the sentence in the introduction about them being static is inaccurate.*
>
> A5. Thanks for pointing this out. We have revised the corresponding descriptions. We actually mean these datasets are not constantly evolving in a timely manner. Sorry for causing the confusion.
>
> ***
> *Q6. The claim in Fig 8, that these are wash trading patterns. Which study has found these?*
>
> A6. Thanks for pointing this out. We define the wash trading patterns by utilizing wash trading’ definition as well as the findings provided in reference [1,2]. Wash trading is an activity where the seller is on both sides of the trade. The more complex the structures, the less frequently they appear. Based on the above information, we first identify wash trades according to the information provided in https://dune.com/qboy29/looksrare-wash-trades, which treats "back-and-forth trades", very high sales and abnormal rewards as wash trades. Then, we sort the wash trades in descending order of their frequency and selected the top five patterns.
>
> [1] von Wachter V, Jensen J R, Regner F, et al. NFT Wash Trading: Quantifying Suspicious Behaviour in NFT markets[C]//Financial Cryptography and Data Security. FC 2022 International Workshops.
>
> [2] Victor F, Weintraud A M. Detecting and quantifying wash trading on decentralized cryptocurrency exchanges[C]//Proceedings of the Web Conference 2021. 2021: 23-32.
>
> ***
> *Q7. There are a few things on the website that are not mentioned in the paper, such as the CSM data. What is it?*
>
> A7. Thanks for pointing this out. CSM stands for Continuous Subgraph Matching, which is one of the downstream tasks that we perform in appendix. Now, we have included all the downstream tasks in the main paper. Meanwhile, we have updated our website (https://livegraphlab.github.io) and enhanced its clarity.

---

> > ### Comment · Reviewer_qE1c · 2023-08-25
> >
> > Thanks for the responses and the changes. I am looking at the wash trade link, and it still seems arcane to me. I am not expecting additional response, but I hope you consider adding some details for the trades on the websites. What transaction patterns are these (visually)?

---

> > > ### Author Response · Authors · 2023-08-25
> > > **Thanks for your reply!**
> > >
> > > Dear Reviewer qE1c,
> > >
> > > Thank you for your valuable suggestion. We have taken your feedback into account and made enhancements to our website by incorporating more trade details and illustrative figures. Here's a toy example that exemplifies pattern 1: *Address A (0x7440e1407f95f33206fb72464a63cd54b2ee6282)* initiated the sale of Azuki token 1215 to *Address B (0xd39e456c22eabf3ab0e58bdeea269927c65f6263)*. Subsequently, Address B sold the token to *Address C (0xeaaeac965449d2426f6f793770b4f3560eeb7c0f)*, and eventually, Address C sold it back to Address A. These transactions happened within half an hour, and interestingly, the token's price surged from 8.98 ETH to 11.99 ETH. Such activity can raise suspicions of wash trading. We have provided similar examples for the remaining patterns on our website (https://livegraphlab.github.io), aiming to provide users with a clearer grasp of these patterns.
> > >
> > > Feel free to reach out if you have any further questions or suggestions.

---

### Official Review · Reviewer_e4Wv · 2023-07-27

**Rating:** 7
**Confidence:** 3

**Strengths:**

The paper proposes a large, open, and publicly available temporal graph dataset for analyzing NFT blockchain systems.
The dataset is constantly evolving in real-time, capturing all interactions with realistic timestamps.
Extensive experiments using Graph machine learning provide valuable insights into NFTs compared to social networks.
The research enriches current graph datasets and offers unique patterns from real-world applications.
The website is well-organized, presenting potential tasks and illustrating real-world impact thoroughly.
The experiments are extensive, informative, and demonstrate the dataset's potential for various applications.


**Additional Feedback:**

None

**Clarity:**

The paper is generally well written, but there are areas where improvements in organization, fluency, and presentation can be made. Specifically, the results analysis section could benefit from being more concise and integrated into the main content rather than being relegated to the Appendix.

Additionally, Table 1 and Table 2 should be placed on the same page where they are discussed to enhance readability. The text in Figure 1 and Figure 2 is too small, and the figures themselves need to be made more robust to provide a better understanding of the context.


**Correctness:**

The claims and methods presented in the paper are overall sound and well-supported.


**Documentation:**

The documentation of data, meta data, data loader API is very well written.


**Ethics:**


The paper acknowledges that the publicly available blockchain data may raise concerns about easier access for users with malicious intent, potentially leading to non-traceable attack vectors. However, since NFT data is already public, privacy concerns in this context are not applicable.


**Limitations:**

While the authors mentioned limitations, more detailed discussions on these aspects and potential negative societal impacts would be beneficial.

**Opportunities For Improvement:**

Writing clarity could be enhanced.
Evaluation experiments may be computationally expensive for a general audience, considering potential out-of-memory issues. A sub-sampled subset with smaller GPUs like 32G v-100 could be proposed as an alternative.
The website and demo need improvement with more informative and straightforward illustrations or examples.


**Relation To Prior Work:**

The paper adequately discusses its differences from previous contributions, especially within the financial domain. However, it could expand the discussions to include other domains and briefly introduce common tasks like anomalous detection from prior work to strengthen its relation to existing research. This would enhance the clarity of the contributions and highlight the uniqueness of the approach.


**Summary And Contributions:**

The paper presents a solid dataset for temporal graph analysis of Non-fungible tokens (NFTs) in blockchain systems. The dataset is open, publicly available, and constantly updated in real-time, providing a complete record of all interactions with realistic timestamps.

The paper's main contributions lie in its extensive experiments using Graph machine learning, which offer valuable insights into NFTs compared to social networks. Furthermore, the dataset enriches the current resources available for graph research.

---

> ### Author Response · Authors · 2023-08-22
> **Response to Reviewer e4Wv**
>
> We sincerely thank the reviewer e4Wv for the careful reading and valuable feedback. We present the following detailed responses to address your concerns:
>
> *Q1. Evaluation experiments may be computationally expensive for a general audience, considering potential out-of-memory issues. A sub-sampled subset with smaller GPUs like 32G v-100 could be proposed as an alternative.*
>
> A1. Thanks for pointing this out. Following your valuable suggestion, we have sampled a subset of data spanning from January 1st 2020 to Aug 31st 2020. This subset contains 88,112 nodes and 203,221 edges. We provide the temporal link prediction results under live update setting as follows:
>
> | Methods | Snapshot | Days | Snapshot | Weeks | Snapshot | Months |
> |:--:|:--:|:--:|:--:|:--:|:--:|:--:|
> | Metrics | AUC | MRR | AUC | MRR | AUC | MRR |
> | Dyngraph2vec | 77.39$\pm$2.04 | 40.29$\pm$5.75 | 75.33$\pm$7.15 | 36.84$\pm$4.97 | 72.03$\pm$2.73 | 34.83$\pm$4.20 |
> | TGCN | 95.49$\pm$0.78 | 62.40$\pm$7.89 | 91.09$\pm$0.46 | 41.69$\pm$7.82 | 86.13$\pm$1.33 | 43.67$\pm$6.23 |
> | EvolveGCN | 84.34$\pm$2.81 | 40.91$\pm$7.96 | 79.13$\pm$0.92 | 40.72$\pm$3.82 | 78.40$\pm$3.33 | 42.86$\pm$3.71 |
> | GCRN-GRU | 91.41$\pm$1.24 | 33.65$\pm$9.48 | 90.78$\pm$2.43 | 42.88$\pm$0.48 | 82.64$\pm$0.22 | 38.99$\pm$0.18 |
> | GCRN-LSTM | 93.97$\pm$1.66 | 41.63$\pm$3.22 | 90.16$\pm$1.83 | 39.20$\pm$7.19 | 83.38$\pm$0.71 | 37.22$\pm$2.63 |
> | DynGEM | 95.67$\pm$0.74 | 49.25$\pm$6.01 | 93.34$\pm$0.98 | 45.41$\pm$5.29 | 90.75$\pm$1.29 | 41.38$\pm$0.61 |
> | Roland-MA | 97.55$\pm$0.59 | 37.18$\pm$6.76 | 94.59$\pm$0.50 | 44.26$\pm$1.26 | 87.69$\pm$2.17 | 37.98$\pm$4.61 |
> | Roland-MLP | 96.38$\pm$1.07 | 60.44$\pm$10.0 | 89.25$\pm$0.89 | 52.32$\pm$3.20 | 87.89$\pm$0.96 | 42.02$\pm$3.37 |
> | Roland-GRU | 96.52$\pm$0.08 | 69.34$\pm$2.68 | 92.83$\pm$0.82 | 52.74$\pm$2.60 | 91.01$\pm$0.77 | 41.59$\pm$2.02 |
>
> Based on the results, we can draw similar conclusions outlined in our paper. Notably, we observe that with the increase in time granularity, the performance of the models exhibits a corresponding decline within this sub-sampled dataset.
>
> ***
> *Q2. The website and demo need improvement with more informative and straightforward illustrations or examples.*
>
> A2. Thanks for pointing this out. We have updated our website (https://livegraphlab.github.io) based on your suggestion. We have included additional illustrations that demonstrate the procedures for collecting the datasets and performing the downstream tasks.
>
> ***
> *Q3. While the authors mentioned limitations, more detailed discussions on these aspects and potential negative societal impacts would be beneficial.*
>
> A3. Thanks for pointing this out. We provide more discussions here. The Live Graph Lab can facilitate researchers from graph community by offering comprehensive blockchain based graphs via an easily accessible manner. Insights from this research can be directly applied to improve the design, security, and user experience of NFT platforms, leading to sustainable growth of the ecosystem. Meanwhile, as the live graph constantly records all the NFT transactions in the Ethereum blockchain, the possibility of encountering malicious activities could become a concern, such as bot transactions or wash trading, etc. It might also cause potential negative societal impacts. Since the dataset consists of complete transactions associated with the wallet addresses, it could enable the tracking of each wallet’s behaviors, habits, and financial activities. This kind of tracking could be exploited for targeted advertising, manipulation, or surveillance. The dataset could also make it possible for malicious actors to analyze the transaction patterns and manipulate the NFT market, which could lead to unfair practices, price manipulation, and market instability.
>
> ***
>
> *Q4. The results analysis section could benefit from being more concise and integrated into the main content rather than being relegated to the Appendix.*
>
> A4. Thanks for pointing this out. We have updated the contents in Section 5 and incorporated all the downstream tasks into the main paper. For more details, please refer to the revised paper. We have utilized blue color to highlight the modifications.
>
> ***
> *Q5. Table 1 and Table 2 should be placed on the same page where they are discussed to enhance readability. The text in Figure 1 and Figure 2 is too small, and the figures themselves need to be made more robust to provide a better understanding of the context.*
>
> A5. Thanks for pointing this out. We have reorganized the Table 1 and Table 2 to appear on page 3. Additionally, we have increased the text size in Figure 1 and Figure 2 to ensure better clarity. For more details, please refer to the revised paper.

---

> ### Author Response · Authors · 2023-08-26
> **A gentle reminder for discussion**
>
> Dear Reviewer e4Wv,
>
> We would like to express our sincere gratitude for the time and effort you have put into reviewing our paper and providing us with valuable feedback. We have tried our best to address your concerns.
>
> We kindly request that you take a few moments to review our rebuttal and let us know if there are any further concerns that we can address. We would appreciate your prompt response.
>
> Thank you once again for your feedback.

---

### Author Response · Authors · 2023-08-22
**Summary of Paper Updates**

We sincerely appreciate the insightful and constructive feedbacks from all the reviewers. We are particularly encouraged that reviewers recognize the contribution of our proposed dataset.

We have addressed all the questions of reviewers in each individual response. According to the suggestions of reviewers, we also conduct additional experiments on a sampled subset and include the results in the revised paper. The modifications in the paper are highlighted in blue. Below is a summary of paper updates.
- We have reorganized the Table 1 and Table 2 to appear on page 3. Additionally, we have increased the text size in Figure 1 and Figure 2 to ensure better clarity.
- We have revisited the content in Section 5 and integrated all the downstream tasks into the main paper.
- In Appendix Section F, we provide additional experimental results on a sampled subset.
- As suggested by the reviewer, we have included the TEA and TET plots in Appendix Section G.

We hope the updated results and our responses have addressed the questions that reviewers have. Please let us know if there are any other questions or suggestions.

Thanks,

Authors of Submission 649

---

### Decision · Program_Chairs · 2023-09-22

**Decision:**

Accept (Poster)

**Comment:**

This paper proposes the concept of live graph lab for temporal graphs and constructed a open, large-scale and temporal graph constructed from NFT dataset. The Live Graph Lab can periodically collect the latest NFT transactions for updating the graph. The constructed graph has rich tasks and properties that could benefit blockchain and graph machine learning research. The paper also conducted comprehensive analysis of the graph and tasks. In summary, the proposed large-scale graph data has great potential for benefiting graph machine learning community.